



# Quantifying pluriannual hydrological memory with Catchment Forgetting Curves

Alban de Lavenne[1,2], Vazken Andréassian[2], Louise Crochemore[1,3], Göran Lindström[1], and
Berit Arheimer[1]

[1]SMHI, Norrköping, Sweden
[2]Université Paris-Saclay, INRAE, UR HYCAR, Antony, France
[3]INRAE, UR RiverLy, Lyon, France

**Correspondence:** Alban de Lavenne (alban.delavenne@inrae.fr)

**Abstract.** This article presents a new approach to quantifying pluriannual hydrological memory, using exclusively streamflow and climate data. The rainfall–runoff relationship is analyzed through the concept of elasticity, focusing on the relation between the annual anomalies of runoff yield and humidity index. We identify *Catchment Forgetting Curves* (CFC) to quantify pluri-annual catchment memory, considering not only the current year's humidity anomaly but also the anomalies of the preceding

years. CFCs are parameterized using a Gamma distribution.

The variability of CFCs is investigated on a set of 158 Swedish and 527 French catchments. As expected, French catchments overlying powerful aquifers exhibit a long memory. In Sweden, the expected effect of the lakes is less clear. Overall, aridity appears to be one of the main drivers of catchment memory in both countries. Our work underlines the need to account for catchment memory in order to produce meaningful and geographically coherent elasticity indices.

## 1 Introduction

### 1.1 Catchment memory

A catchment receives precipitation from the atmosphere, a water amount that is either stored in soils, biota, snow/glaciers, lakes/wetlands, and aquifers, or returned to the atmosphere (evaporated), exported by the river (as streamflow), or exported to regional aquifers (as intercatchment groundwater flow). The relative distribution between these fluxes depends not only on the

physical characteristics of the catchment, but also on the recent climatic sequence: The response of a catchment to incoming precipitation depends largely on its *wetness* (i.e., on the more or less saturated state of soils and wetlands within the catchment). One can thus talk of *catchment memory*, in that in its reaction to the incoming precipitation, a catchment remembers the past to some extent.

The objective of this paper is to characterize catchment memory, in order to understand the time during which the past

climatic sequence will affect catchment response. We wish here to follow the experimental psychologist Ebbinghaus (1885) in proposing *forgetting curves*, which will describe the decline of memory in time. To make this discussion of a complex matter simple, we start with a first-order simplifying assumption: We hypothesize that a catchment may have both a short-term and





a long-term memory (see e.g., Risbey and Entekhabi, 1996; McDonnell, 2017); we consider the short-term memory to be
seasonal, and will not address it in this paper in order to focus on the long-term (pluriannual) memory effects.

In what follows, we will consider catchment memory from the point of view of annual precipitation yield (i.e., the ratio
of annual discharge to annual precipitation) and will research the different climatic factors explaining its variability. We aim
in particular to assess the cumulative effect of wet and dry years, i.e., how successions of relatively wet or dry years within a
climatic sequence affect the rainfall yield over subsequent years. While catchment memory is obviously a function of catchment
storage capacity (in groundwater aquifers, wetlands, lakes or glaciers), the originality of this paper will be in the quantification

of *forgetting curves* at catchment scale: We will identify them from annual data series and, only then, attempt a physical
interpretation.

## 1.2   Catchment memory vs. water age

A distinction that we believe is necessary from the onset is the one between catchment memory and water age. Indeed, because
catchment memory reflects a temporal aspect, the difference between these two notions might look tenuous, and it is important

to clarify it:

**water age**  describes the time that the water takes to travel through the catchment. It can describe the actual age of the water
storage (residence time distribution) or the age of the water when it reaches the outlet (travel time distribution). This
is a primary focus when it comes to water quality modeling, where flow paths and travel time have to be understood
for tracking nutrients within the catchment (Hrachowitz et al., 2016). As summarized in the recent review by Sprenger

et al. (2019), these investigations will generally rely on tracers and on a physical understanding that would explain how
catchment storage is sampled by different hydrological processes to generate streamflow. Because we do not use any
tracers in this study, we cannot check any hypothesis about water age and we will not discuss this topic further;

**catchment memory,**  as defined in this paper, describes the period of time during which we manage to detect a significant
dependency between two signals: the past climatic inputs and the current ability of the catchment to transform precipita-

tion into river flow. The scientific literature sometimes addresses catchment memory also through "flow persistence" (see
e.g., Svensson, 2015; Quinn et al., 2021) or "flow predictability" (see e.g., Bierkens and van Beek, 2009; van Dijk et al.,
2013). Compared with a description of water age, the ambition to physically understand the system is more limited and
restricted to explaining catchment behavior from the perspective of an operational flow prediction model. Our study is
thus more in line with conceptual modeling, where we focus more on the concept of celerity (pressure wave propagation)

rather than on velocity (mass flux of water) to describe the hydrological response (McDonnell and Beven, 2014).

## 1.3   How to describe catchment memory?

There is a broad literature dealing with catchment memory. Among the authors who have discussed related topics in the past,
the contribution of Hurst (1951) is one of the earliest (see also the review by O'Connell et al., 2016). He was followed by many





hydrologists who studied the auto-regressive properties of annual flows (e.g. Lins, 1985; Montanari et al., 1997; Vogel et al.,
1998; Rao and Bhattacharya, 1999; Wang et al., 2007; Mudelsee, 2007; Szolgayova et al., 2013).

Spectral analysis can be used to provide insight into catchment memory. It is regularly used for stream chemistry (see e.g.
Kirchner et al., 2000), in order to understand travel time distributions. Despite being out of the scope of this study (see section
1.2), these spectral analyses generally highlight that catchments do not exhibit a particular flushing time of a contaminant, but
instead a rapid flush followed by a low level of contamination that could be surprisingly long.

Simple correlations are a common method for quantifying the dependence on the past. Nippgen et al. (2016) studied the lag
correlation between precipitation and runoff ratio, from monthly to annual time steps, and demonstrated that the precipitation
of the previous year was equally correlated to the year's runoff ratio in five North Carolina catchments studied. Iliopoulou
et al. (2019) computed a lagged seasonal correlation between selected river flow signatures and the average river flow in the
antecedent months. They found higher correlations with low-flow signatures than with high-flow signatures.

Gharari and Razavi (2018) analyzed the memory of hydrological systems from the point of view of hysteretic systems, using
"path-dependent systems" as a synonym of "systems with memory", and precising that unlike a system in which the future
state depends only upon its present state and forcings, the future of a path-dependent system depends on the sequence of states
preceding the present state.

Catchment memory can also be approached by quantifying water storage. Creutzfeldt et al. (2012) used a 10-year time series
of high-precision gravimetric measurements to follow the evolution of catchment-scale water storage, as well as the long-term
recovery of a particularly strong drought event in 2003, and found that the catchment remembers the event over several years.
Orth and Seneviratne (2013) explain streamflow and evapotranspiration memory as a propagation of soil moisture memory.
Instead of quantifying memory using a lag correlation, they proposed calculating the mean time required to recover from
anomalous conditions. This enabled them to highlight a longer memory for the more extreme anomalies.

Multi-year memory can also be analyzed through the residuals of annual water balance. Trask et al. (2017) proposed different
statistical techniques to account for these residuals of the previous years. It enables one to improve each annual water balance
evaluation and to account for inter-annual changes of water storage.

Any hydrological model must, in one way or another, conceptualize the hydrological memory by parameters that govern the
behavior of model states. Kratzert et al. (2019) proposed a data-driven approach using Long Short-Term Memory networks
(LSTMs) to simulate discharge. LSTM is a class of neural network where each cell has a memory coming from long-term
dependencies between input and output features. This memory conceptualization is quite similar to a state in a hydrologi-
cal model. Memory can also be derived from a digital filter applied on the hydrograph. Pelletier and Andréassian (2020b)
introduced a memory-based approach for determining the parameters of a conceptual baseflow separation method.

In summary, it appears that most existing methods aiming to analyze memory either summarize the memory by a single
value and/or provide an index that cannot be directly interpreted as duration. In this study, we seek to describe memory in the
form of duration but also to understand how this memory fades over time, i.e., how the catchment forgets.





### 1.4 Why describe catchment memory?

The concept of catchment memory is used with intra-annual objectives to qualify the predictability linked with initial hydrological states (e.g., Svensson, 2015; Bierkens and van Beek, 2009; van Dijk et al., 2013; Quinn et al., 2021). Studies have aimed

to weigh the predictability linked with the past, and conveyed by hydrological states (or catchment memory), in relation to the predictability linked with future rainfall and temperatures for predicting discharge (Wood et al., 2016; Arnal et al., 2017). A critical time horizon in these studies, rather than the time when initial states no longer influence future outcomes, is the time when the predictability linked with catchment states is outweighed by future conditions (e.g., Yossef et al., 2013; Shukla et al., 2013). So far, short-term memory (seasonal storage) has often been the main focus in these studies. Pluriannual memory has

been explicitly distinguished only recently with advances in decadal forecasting (Yuan and Zhu, 2018).

Catchment memory also has a long history when it comes to water quality modeling or tracer analysis, as past pollution inputs can influence water quality in rivers for several years or decades, creating a legacy that is often difficult to estimate (e.g., Hrachowitz et al., 2015; Van Meter et al., 2016). The time lag to achieve water quality goals, such as nitrogen reduction, thus have to be efficiently captured by the models (e.g., Ilampooranan et al., 2019). Landscape analysis based on water and

substance retention in various storages during the flow path from sources to the sea is important for judging the effects of remedial measures (e.g., Arheimer and Wittgren, 2002; Arheimer et al., 2015). However, as discussed above, this definition of memory is more in line with the studies of travel time, which is not the direction adopted in this paper (see section 1.2).

Catchment memory is sometimes seen as a way to understand the nonlinearity of streamflow response to precipitation (Risbey and Entekhabi, 1996), and some authors see a better characterization of catchment memory as essential for model

structure improvement. For instance, Fowler et al. (2020) analyzed the slow dynamics observed in catchments and argue that streamflow may depend not only on the climatic conditions of the preceding weeks or months, but also on past years or even decades. They consider that hydrological models often lack an explicit description of these long-term effects. Grigg and Hughes (2018) analyzed memory effects caused by multidecadal changes in catchment groundwater storage and showed that the GR4J model requires a complexification to account for these effects. The modification they propose is shown to be coherent with

groundwater observations.

Catchment memory is also studied to understand how catchments recover from climatic extremes, such as multiyear droughts (Creutzfeldt et al., 2012; Hughes et al., 2012; Yang et al., 2017). Merz et al. (2016) hypothesized that catchment memory, along with intra- to inter-annual climate variability, could be responsible for temporal clustering of floods in Germany.

### 1.5 What drives catchment memory?

Several studies have linked catchment memory to catchment size, generally showing that memory increases with catchment size (see e.g., Mudelsee, 2007; Hirpa et al., 2010; Szolgayova et al., 2013; Iliopoulou et al., 2019). This is usually explained by the increase of storage in larger catchments. Mudelsee (2007) also explained the Hurst index through catchment geomorphology and the cascade produced by spatial aggregation along the river network.





Other physical descriptors, such as soil and geology, are regularly used to provide a physical explanation of catchment
memory. Merz et al. (2016) noted that catchments with deep soils and/or saprolite zones and powerful aquifers have a higher
catchment state persistence and thus a stronger memory. Orth and Seneviratne (2013) showed that soil moisture to some extent
serves as an upper bound for streamflow and evapotranspiration memory.

Some authors also assessed whether memory can be identified through hydrological signatures. Szolgayova et al. (2013)
found that the Hurst index generally increases with mean discharge, but decreases with specific discharge. Memory is often
related to groundwater storage through indicators such as baseflow index (Grigg and Hughes, 2018; Fowler et al., 2020;
Pelletier and Andréassian, 2020a). Several papers have shown that a long seasonal predictability is often correlated with the
importance of baseflow (Harrigan et al., 2018; Lopez et al., 2021; Iliopoulou et al., 2019). Tomasella et al. (2008) described
a strong memory effect in a small 6-km$^2$ Amazonian catchment that impacts the hydrological response of the catchment well
beyond the time span of the seasonal climatic anomalies. The authors attribute this memory to storage in the groundwater
and unsaturated zone, and warn against the impact of this memory effect on the closure of water balance by models. Similar
phenomena are found close to the Sahara Desert where aquifers may interfere with the surface water of the catchments in areas
of certain geology (Andersson et al., 2017). However, when it comes to short-term memory, Lo and Famiglietti (2010) showed
that the presence of a groundwater aquifer can either increase or decrease land surface hydrological memory and this depends
on the depth of the water table.

Memory is also often related to different dryness indices. Szolgayova et al. (2013) found that memory increases with mean
air temperature. In dry regions, the past generally weighs more on the predictability of seasonal forecasting than it does in
wet regions (Harrigan et al., 2018; Lopez et al., 2021). Iliopoulou et al. (2019) showed that memory decreases with increased
wetness conditions.

Humans also affect the memory of hydrological systems: One can cite, for instance, the Sahelian paradox where runoff has
increased since the 1970s, despite decreases and sustained low levels of rainfall (e.g., Amogu et al., 2010; Descroix et al.,
2009) probably due to land degradation and soil crusting resulting in Hortonian overland flow instead of infiltration. Similarly,
tile drainage of arable land has had large effects on soil storage in Sweden, providing a shorter catchment memory (Andersson
and Arheimer, 2003). When building reservoirs, on the contrary, the memory is extended.

### 1.6 Scope of the paper

A concept that seemed particularly handy to describe synthetically the relationship between two variables is *elasticity*. Taken
from economics (Marshall, 1890), it has been applied in hydrology to describe the sensitivity of the changes in streamflow
to changes in a climate input variable without requiring a rainfall–runoff model (Schaake and Liu, 1989; Andréassian et al.,
2016). To our knowledge, it has never been applied with explicit consideration of catchment memory. The goal of our paper is
threefold:

1. To present a method, based on the concept of elasticity, that not only can provide an index relevant to catchment memory,
    but can also characterize its dynamic in a manner analogous to a *forgetting curve* (Ebbinghaus, 1885);





2. To disentangle catchment memory and catchment elasticity;

3. To provide some physical indicators of the main drivers of memory and elasticity.

## 2   Material and Methods

### 2.1   Catchment data

A total of 685 catchments are used in this study: 527 French catchments and 158 Swedish catchments.

In France, discharge series ($Q$) were extracted at a daily time step from the French Hydro database (Leleu et al., 2014). Only catchments that are not regulated according to this database were selected. Corresponding catchment areas vary between 5 km$^2$ and 26,900 km$^2$. Precipitation ($P$) and temperature ($T$) data were extracted from the SAFRAN atmospheric reanalysis produced by Météo-France (8x8 km and averaged over the catchment upstream areas; Vidal et al., 2010).

In Sweden, discharge series were extracted at a daily time step from the SMHI database. Only stations having less than 5% of their area regulated are used (information extracted from www.smhi.se). Catchment areas vary between 1 km$^2$ and 14,400 km$^2$. Precipitation and temperature data were extracted at a daily time step from the PTHBV database (4x4 km; Johansson, 2002).

For all stations, potential evaporation ($E_0$) was estimated following Oudin et al. (2005). Daily hydroclimatic data were aggregated at the annual scale for the purpose of this study, with the hydrological year starting on October $1^{st}$. We accepted a maximum of 10% of missing data per year. All 685 stations also respect a period of at least 10 years of hydroclimatic data without any gaps in order to be able to capture long-term memory. In the end, between 23 and 59 gauged years were available for our catchments.

These time series are also used to build five hydroclimatic descriptors for each catchment with the average of the annual values of $Q$ and $P$, $E_0$, $Q/P$ and $P/E_0$. In addition, the percentage of the watershed area covered by lakes is used as a supplementary descriptor. This information is provided by SMHI for Swedish catchments (extracted from www.smhi.se). For France, the Lake Water Bodies according to the Water Framework Directive is used (extracted from geo.data.gouv.fr). Finally, the contribution of groundwater is assessed by a base flow index (BFI) calculated according to the work of Pelletier and Andréassian (2020a) with the associated "baseflow" R package (Pelletier et al., 2021).

### 2.2   Discrete memory conceptualization

Our memory conceptualization starts discretely, i.e., on a year-by-year basis, and uses the concept of catchment elasticity (Schaake and Liu, 1989; Andréassian et al., 2016). Elasticity describes the sensitivity of the changes in streamflow related to changes in a climate input variable. More precisely for this study, we focus on the sensitivity of the changes in runoff yield ($Y = Q/P$) related to changes in humidity index ($H = P/E_0$) computed at the annual time step, as described by equation 1.

$$\frac{\delta Y}{\overline{Y}} = \varepsilon_1 \frac{\delta H}{\overline{H}}$$

(1)





where $\overline{Y}$ and $\overline{H}$ are long-term average values of catchment yield and humidity indices, respectively, the operator $\delta$ indicates the difference between a dated and a long-term average value, and $\varepsilon_1$ is the elasticity index.

In order to investigate memory effects, we need to add a temporal dimension $i$ to the traditional relation defined in equation 1: Instead of trying to explain the yield anomaly of year $i$ from the climatic anomaly of the same year $i$, we allow for the use of several past climatic anomalies. The influence of each past anomaly is quantified by an additional parameter $\omega_i$, with $i$ varying from 0 (the current year) to $n$ preceding years (fixed to $n = 5$). By estimating the different values of $\omega$ over the past years, we will be able to construct the (discrete) *Catchment Forgetting Curve* (CFC): It describes how quickly a catchment forgets past anomalies and when it starts to behave independently from past years' events. The elasticity index $\varepsilon_2$ is still quantified (as in equation 1) and distinguished from catchment memory $\omega$.

$$\frac{\delta Y_0}{\overline{Y}} = \varepsilon_2 \sum_{i=0}^{n} \left( \omega_i \cdot \frac{\delta H_i}{\overline{H}} \right)$$
$$\text{with} \sum_{i=0}^{n} \omega_i = 1 \tag{2}$$

Graphically, the memory effect can be visualized by a series of plots showing the runoff yield anomaly in the ordinate as a function of the climate anomaly of the preceding years in the abscissa (Figure 1, top). Figure 1 shows a real example with a rather peculiar behavior: a catchment where past climatic anomalies are much better related to the past yield anomaly ($i \in \{1, 2, 3\}$) than the current year anomaly ($i = 0$).

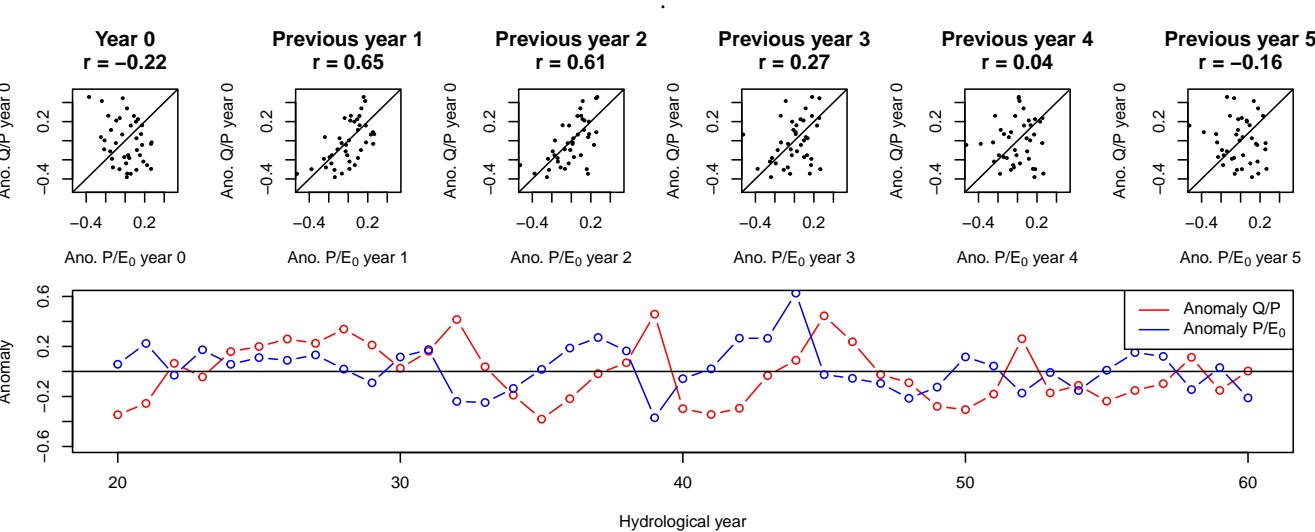

**Figure 1.** Elasticity analysis example on the *Petit Thérain* River ($212\,\mathrm{km^2}$), France. The effect of a climatic anomaly $P/E_0$ on the hydrology ($Q/P$) only starts after 1 year, before slowly decreasing ($r$ - Pearson correlation values).





### 2.3 Continuous memory conceptualization

The parameterization of the discrete CFC (as shown in equation 2) requires the calibration of $n+1$ values of $\omega$. In order to limit the number of parameters and to avoid having to calibrate each $\omega$ value independently, we hypothesized (after many attempts that we cannot report here) that the CFC can be described by a Gamma distribution. This assumption is not uncommon among studies that focus on describing transit time distribution (see for instance: Kirchner et al., 2000, 2001; Dunn et al., 2010; Hrachowitz et al., 2010; Godsey et al., 2010; Tetzlaff et al., 2011; Heidbüchel et al., 2012; Berghuijs and Kirchner, 2017). The choice for a skewed distribution was also driven by visualizing all the plots obtained for our catchment set (see, e.g., Figure 1): We found that a simple exponential parameterization would not be enough as it does not allow lags. Gamma distribution (Equation 3) requires the calibration of two parameters: a shape parameter $\alpha$ and a scale parameter $\beta$:

$$\omega(i) = \frac{i^{\alpha-1}}{\beta^\alpha \Gamma(\alpha)} e^{-\frac{i}{\beta}} \qquad (3)$$

where $\Gamma(\alpha)$ is the Gamma function evaluated at $\alpha$.

For the sake of simplicity, before fitting a Gamma distribution, we first fit a simple annual elasticity model (a zero-memory model), and use a statistical significance test (Student's $t$-Test with $pvalue < 0.01$), to decide whether equation 2 improves significantly on equation 1. If the improvement is not significant, we conclude on the absence of pluriannual memory for that catchment, and keep the simplest representation (that of equation 1). Parameters of both equations were calibrated using particle swarm optimization (PSO) through the hydroPSO R package (Zambrano-Bigiarini and Rojas, 2013).

### 2.4 Characterization of the continuous CFC

In order to quantify catchment memory, we extract two characteristic times from the continuous distribution (Figure 2). Firstly, we extract the time when the Gamma distribution is at its maximum value. This allows us to describe a possible lag between the climatic anomaly and the main resulting hydrological anomaly (called $t_p$). In addition, to describe the speed of memory loss, we extracted the time when the cumulative distribution reaches 75% (called $t_{75}$), but any other percentage could be easily extracted in a similar way.



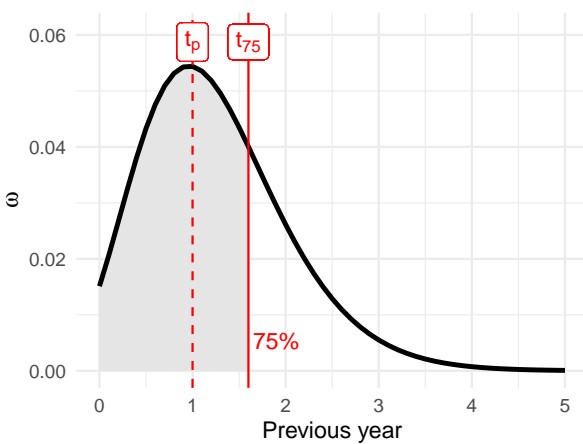

**Figure 2.** Extraction of two characteristic values from an example of a Gamma distribution.

## 3 Results and discussion

### 3.1 Is pluriannual memory a rare phenomenon?

Approximately 80% of the Swedish catchments and 89% of the French catchments showed no significant pluriannual memory (*significant* in terms of the aforementioned Student's *t*-Test): This shows that pluriannual catchment memory is neither common nor very uncommon. We present in Figure 3 the CFCs identified for the Swedish and French catchments separately. In Sweden, many of the catchments with pluriannual memory exhibit a lag of 1 year (Figure 3). This lag also appears in France where it sometimes reaches values of up to 3 years.





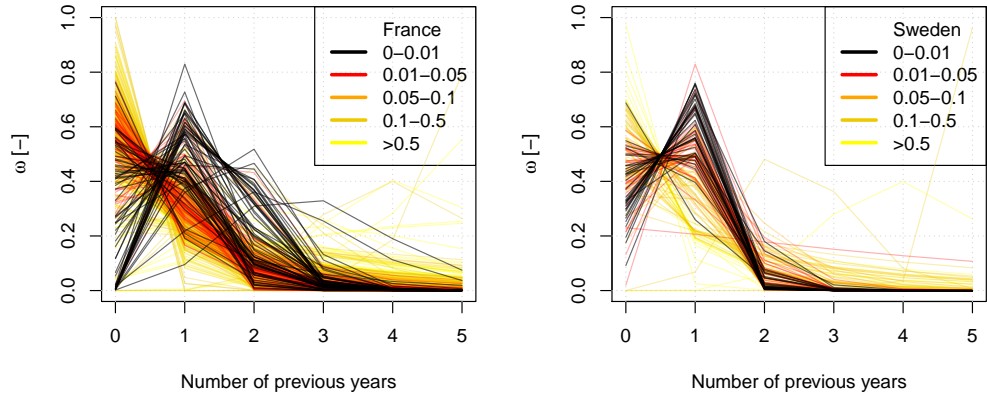

**Figure 3.** Catchment Forgetting Curves identified for the catchments on our French (left) and Swedish (right) dataset. The black lines represent the catchments where the pluriannual CFC is very significant, and the color gradient represents, for the rest of the catchments, the p-value of the $t$-Test.

In France, the weight of year 0 ($\omega_0$) can frequently be equal to 0 (Figure 3): This lag effect means that a climatic anomaly will not have an impact until the next year (as in the example presented in Figure 1). By contrast, Sweden rarely shows such an extreme temporal disconnection, and the climatic anomaly of year 0 is most of the time already affecting catchment yield during the current year.

In France, the calibration of the CFC sometimes yields a slow decrease, from year 0 to year 5, without any lag. But this shape does not really appear in Sweden, where the decrease of the memory is usually fast, with most $\omega$ values becoming negligible already after 2 or 3 years (Figure 3). In France, the $\omega$ values become negligible after 4 or 5 years (which is why we retained 5 years as maximum duration for the CFC).

### 3.2 Where do catchments exhibit a pluriannual memory?

A spatial analysis of the catchments with significant pluriannual memory allows us to already identify some spatial patterns (Figure 4). In France, the Paris basin with its large chalk aquifer is the region where the most significant pluriannual memory exists. In the rest of France, pluriannual memory is generally not significant. In Sweden, long-term memory is mainly detected in the south of the country. Some hydroclimatic characteristics of these regions could explain these spatial patterns (see section 3.3 below).



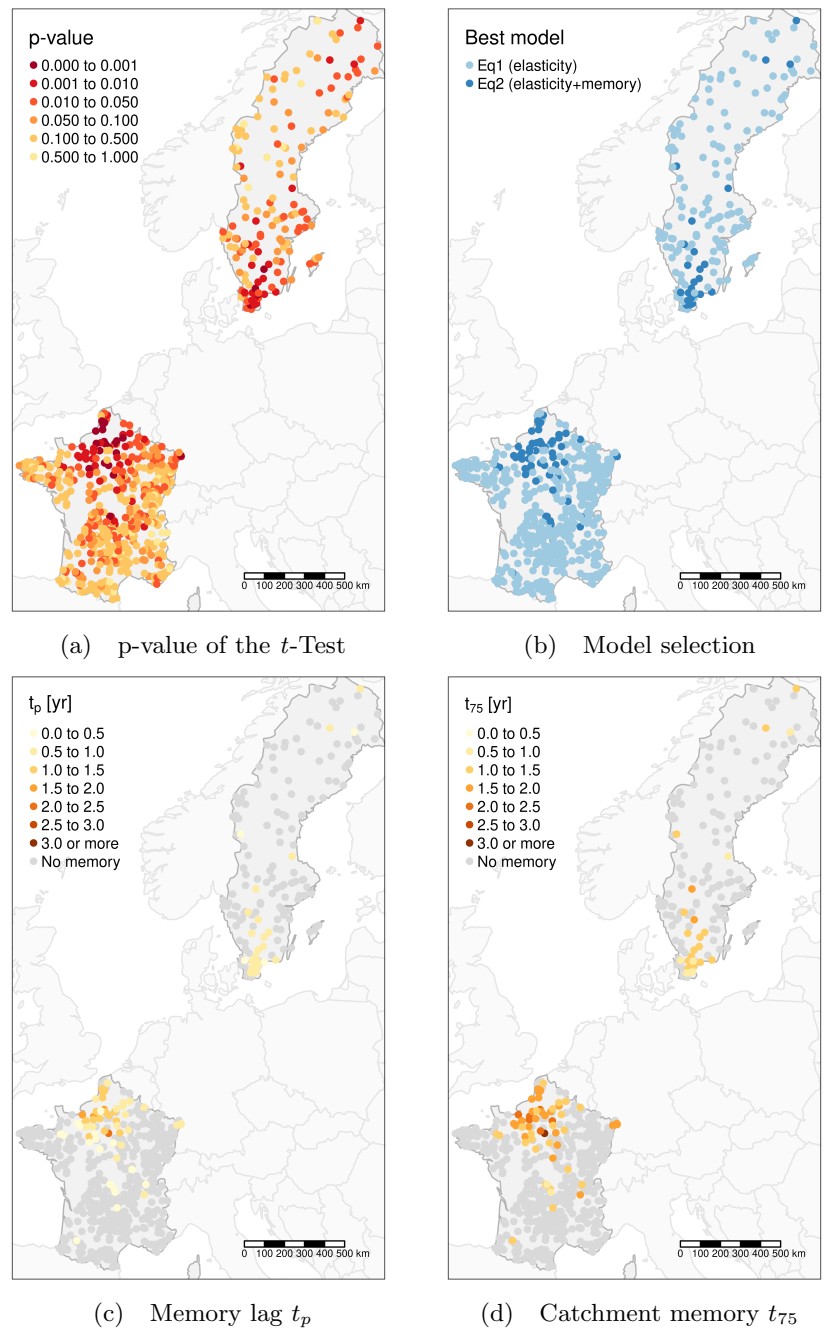

**Figure 4.** Spatial distribution of catchment memory over France and Sweden. Note that the model describing pluriannual catchment memory (equation 2) is used only when the p-value of the $t$-Test is below 0.01. Memory values are extracted from the Gamma distribution (see section 2.4): $t_p$ is the time when the Gamma density function is maximum, and $t_{75}$ is the time when the cumulative distribution reaches 75%.



By extracting the quantile 75% of the cumulated Gamma distribution ($t_{75}$ which we use to characterize the CFC, see section

2.4), it is possible to quantify the duration of catchment memory. For these pluriannual memory catchments, $t_{75}$ is often

between 2 and 3 years in France, whereas it never exceeds 2 years in Sweden (Figure 4).

### 3.3   Can pluriannual memory be explained by hydroclimatic descriptors?

Figure 5 links catchment memory ($t_{75}$) to a few hydroclimatic characteristics commonly identified as the main drivers in the

literature. If larger catchments tend to have larger memory in France, this trend is not confirmed in Sweden.

For both countries, the memory increases with aridity (as characterized by either lower discharge and precipitation, lower

$Q/P$ or lower $P/E_0$). However, the effect of potential evaporation does not appear clearly. This can be understood from the fact

that catchments under humid climate are more likely to respond quickly to climatic inputs, whereas the hydrological behavior

under less humid climate is more variable and linked to the dynamics of long-term water storage.

In France, a higher baseflow index (BFI) clearly identifies catchments with a longer memory. This confirms the predominant

role of powerful aquifers in catchment memory. It also corroborates the spatial analysis of Figure 4, where long memory is

mainly located within the Paris basin where groundwater contributes significantly to total discharge. For France, this spatial

organization is thus very consistent with the memory estimates of Pelletier and Andréassian (2020a).

In Sweden, the percentage of the catchment area covered by lakes (lake cover, Figure 5) does not indicate a longer memory

for catchments with larger lake cover. Compared with France, much of Sweden has thinner soils, which may account for a

lower storage capacity and thus a shorter memory. Hydroclimatic characteristics with long memory in Sweden are consistent

with catchments having higher seasonal prediction potential identified by Lopez et al. (2021) such as a high BFI and a low

amount of precipitation.





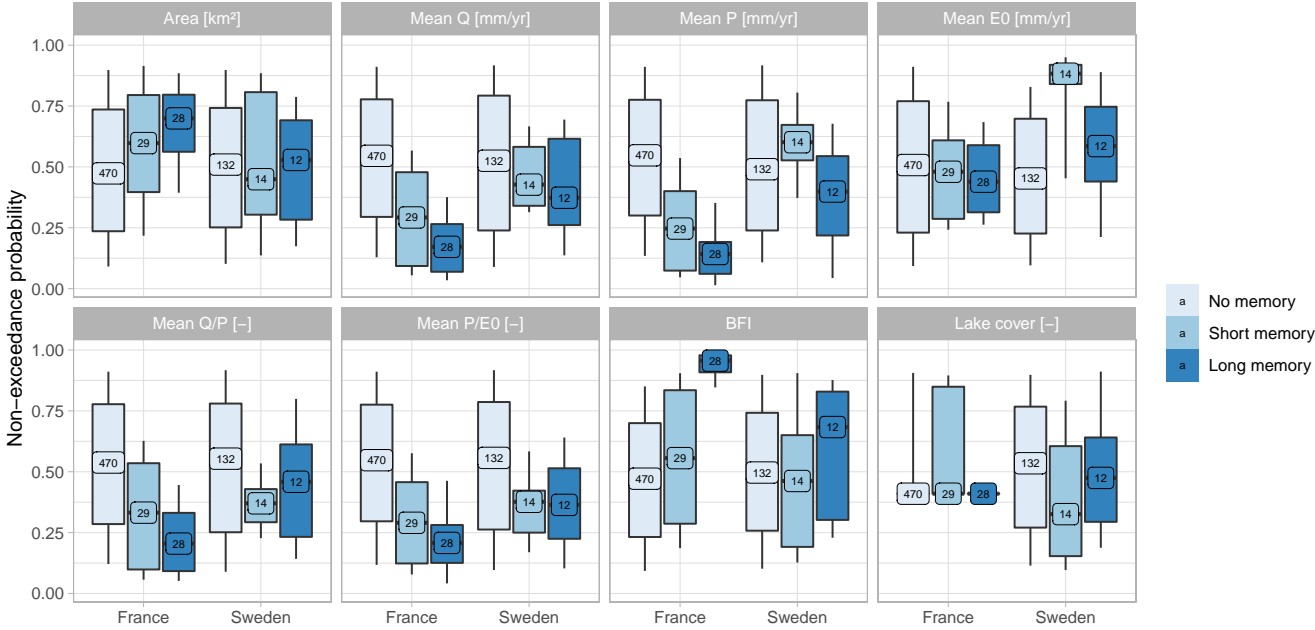

**Figure 5.** Distribution of hydroclimatic characteristics according to three classes of memory (described by $t_{75}$). The first class corresponds to catchments with no significant pluriannual memory, the remaining catchments are split into two groups (shorter memory and longer memory) according to the median value of their memory. The numbers within each boxplot describe the number of catchments. The boxes are delimited by quantiles 0.25 and 0.75; whiskers by quantiles 0.1 and 0.9.

### 3.4 What do we miss in catchment elasticity analysis when not accounting for pluriannual memory?

Equations 1 and 2 both quantify the elasticity $\varepsilon$ in the relation between $Q/P$ and $P/E_0$, but equation 1 does not account

for pluriannual memory effects, while equation 2 does. Figure 6 compares these two elasticity indices and highlights that the elasticity of equation 2 is always higher than the elasticity obtained with equation 1 and generally slightly exceeds the value of 1. Thus, the lower elasticity values obtained with equation 1 are very likely an artifact, the mark of a pluriannual memory-related buffer effect: Elasticity values $\varepsilon_1$ are lower than $\varepsilon_2$ simply to account for the smoothing introduced by catchment memory. When past climatic anomalies are explicitly considered, this buffer effect becomes unnecessary.

By considering catchment memory, the elasticity values are thus changed. However, no strong relations were found between elasticity values and memory values (see Appendix A).





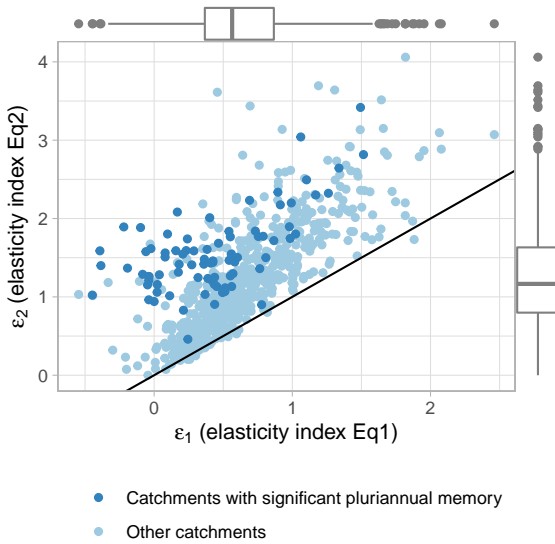

● Catchments with significant pluriannual memory

● Other catchments

**Figure 6.** Comparison of elasticity indices found when ignoring pluriannual memory effects (equation 1) and when explicitly accounting for them (equation 2) for all catchments (France and Sweden).

### 3.5 Can elasticity values be explained by hydroclimatic descriptors?

Similarly to catchment memory, we can try to link elasticity indices to some classic hydroclimatic characteristics. Figure 7 illustrates these relations for the elasticity indices of equation 2 (i.e., the equation accounting explicitly for the memory effect).

The relation between elasticity and catchment area is inverted between France and Sweden: In France, large catchments have larger elasticity, but in Sweden large elasticity is observed in smaller catchments. Our conclusion is that catchment size is not a first-order determining factor of memory and elasticity, and this likely reflects some more regional relation between catchment size and hydrology: For instance, specific discharge tends to increase with catchment size in Sweden, whereas it decreases in France (not shown here).

The stronger trends are found between catchment aridity/humidity and elasticity indices. Similarly to catchment memory (Figure 5), elasticity increases with aridity-related indicators: higher aridity index $E_0/P$ (and lower humidity index $P/E_0$), lower discharge and precipitation, higher $E_0$, lower $Q/P$. This suggests that water-limited catchments not only have a longer memory, but that their hydrological behavior is also more sensitive to climatic inputs. The low values of elasticity in wet areas can also be explained by the fact that the runoff yield, although generally higher, is less variable, thus leading to a lower slope

in the $Q/P$ versus $P/E_0$ relationship (Figure 8).

Higher BFI values coincide with lower elasticity values in both France and Sweden. This suggests that a large contribution of groundwater attenuates the sensitivity to climatic anomalies: Even though these catchments have a longer memory of climatic anomalies, the impact of these anomalies is spread out over the years. In Sweden, lakes also smooth the effect of climatic anomalies, but not as much as humid conditions or large baseflow contributions.





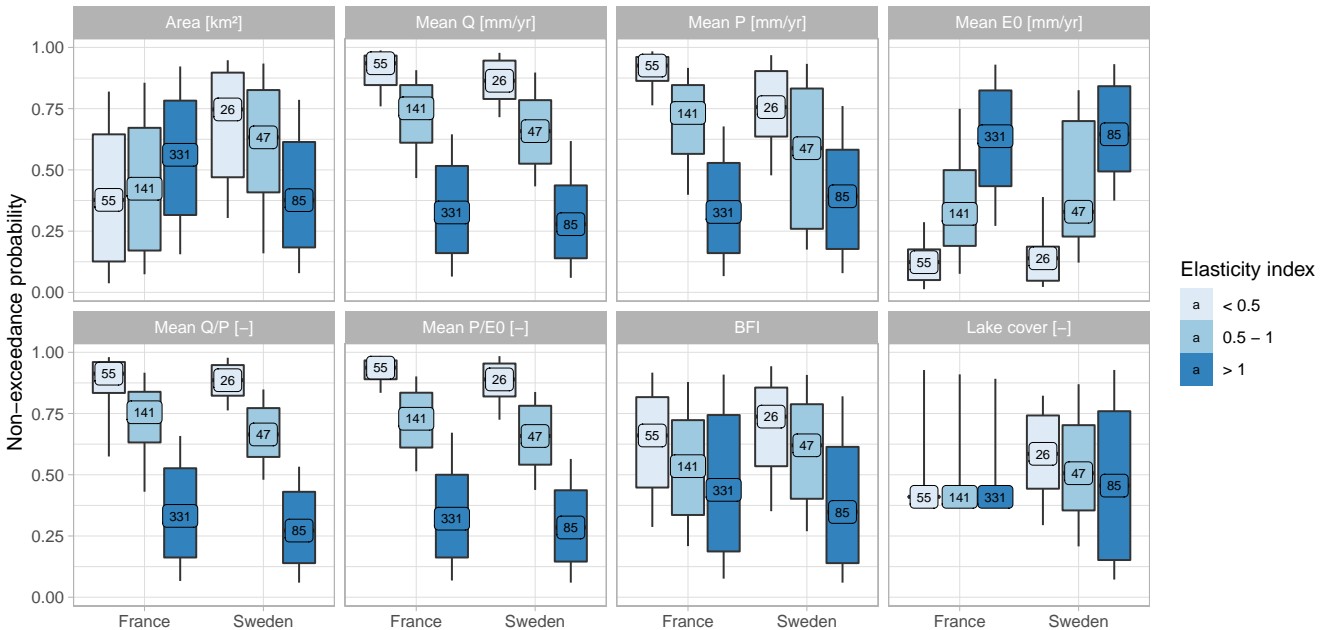

**Figure 7.** Distribution of hydroclimatic characteristics according to three classes of the elasticity index ($\varepsilon_2$) in equation 2. The numbers within each boxplot describe the number of catchments. The boxes are delimited by quantiles 0.25 and 0.75; whiskers by quantiles 0.1 and 0.9.

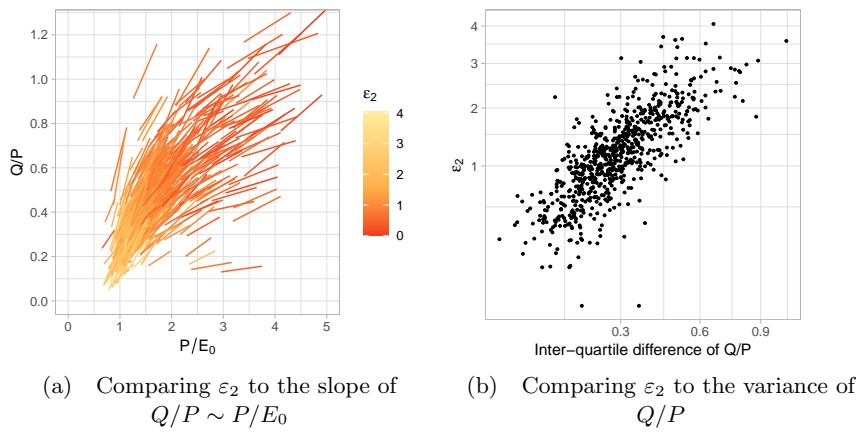

(a) Comparing $\varepsilon_2$ to the slope of $Q/P \sim P/E_0$

(b) Comparing $\varepsilon_2$ to the variance of $Q/P$

**Figure 8.** Comparison of the elasticity index ($\varepsilon_2$, equation 2) with the slope in the linear relation between $Q/P$ and $P/E_0$ (a), and with inter-quartile differences of $Q/P$ values (b). Slopes can be visualized for each catchment through the segments. Segments are drawn between the two points defined by the first and third quartile of these two ratios. A square root transformation is applied on both axes.





The elasticity is also quite well structured in space (Figure 9): This spatial organization reflects the climatic conditions of
each region, as already described by Figure 7. The two elasticity indices ($\varepsilon_1$ and $\varepsilon_2$) generally have the same spatial patterns,
except for the catchments with a significant pluriannual memory: The very low elasticity values that they obtained with equation
1 (that can even be negative) were due to the impossibility to link correctly $Q/P$ and $P/E_0$ without considering the memory
effect. Because it considers explicitly catchment memory, equation 2 yields elasticity values that are more coherent in space
(and avoids the negative values that indicated a lack of hydrological understanding).

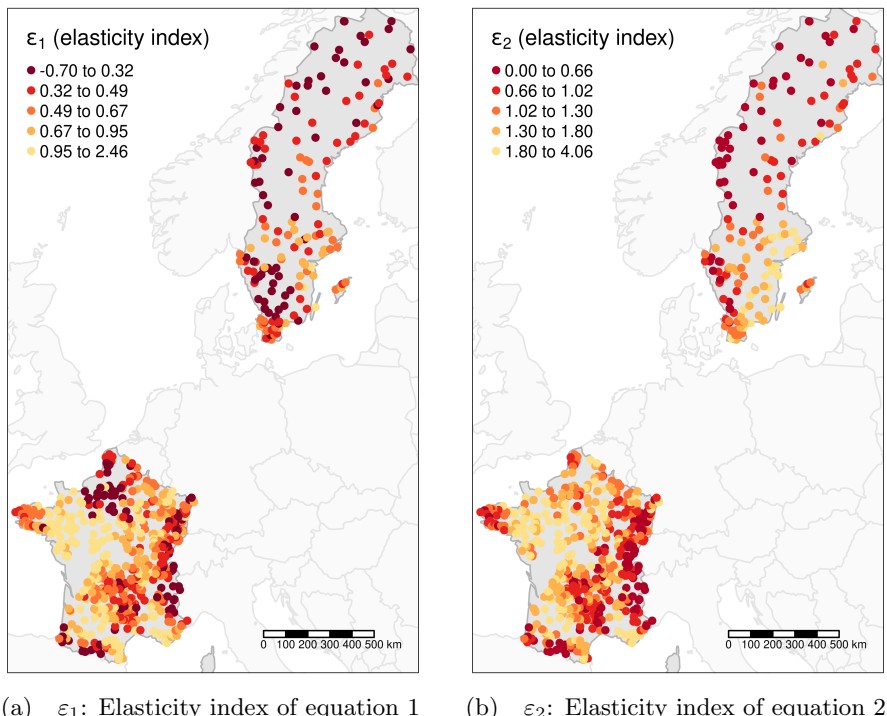

(a)   $\varepsilon_1$: Elasticity index of equation 1    (b)   $\varepsilon_2$: Elasticity index of equation 2

**Figure 9.** Comparison of elasticity index values obtained by optimizing equation 1 (which does not account for memory effects) and equation 2 (which does account for memory effects). The classes are delimited by quantile values to make a relative comparison of spatial patterns easier.

## 4   Conclusions

### 4.1   Synthesis

In this article, we proposed a new approach to quantifying catchment pluriannual memory, requiring only the knowledge of
annual discharge data and climatic inputs. Catchment memory is conceptualized in the form of a *Forgetting Curve* (Ebbinghaus,
1885), characterizing how rapidly a catchment forgets past climatic inputs.





The rainfall–runoff relationship is analyzed through the concept of elasticity, linking the anomalies of runoff yield ($Q/P$) to the anomalies of humidity index ($P/E_0$). In this work, we added a new dimension to the elasticity concept by considering past anomalies and by weighting these anomalies with a Gamma distribution, which gives the shape of the CFC. Last, memory is characterized by the 75% quantile of the fitted Gamma distribution.

As expected, catchments with significant pluriannual memory are dominated by groundwater in France. In Sweden, the expected effect of the lakes does not appear clearly. Instead, the aridity index appears to be one of the main drivers of catchment memory in both countries. Catchment area, often referred to in the literature, does not seem to play a first-order role. The elasticity indices were also well related to aridity, with humid catchments showing lower elasticity. We show that not accounting for pluriannual memory may yield elasticity indices with erratic values (that can even be negative). Introduction of the memory

component produces a much more spatially coherent organization of the elasticity.

## 4.2 Limits

Our methodology relies on a simplifying assumption where short-term and long-term memories are distinguished. It was thus tempting to quote Klemeš et al. (1981), who, at the end of their paper (where they discussed short- and long-memory models), wrote:

*As a scientific hypothesis about streamflow series behavior, neither the short-memory nor the long-memory model can be falsified on the basis of historic flow records of a typical length. Hydrologically, they thus both remain, in Popper's sense, within the realm of metaphysics, and the choice between them is a matter of value judgement. The only arguments that can be advanced for either of them are operational and subjective: Occam's razor and lack of hard evidence to the contrary for short-memory models, hedging against a suspected possibility of a slightly*

*higher risk for the long-memory ones.*

In our case, we would argue that the behavior described in this paper contributes some "hard evidence" on the behavior of hydrological systems, without any unneeded complexity. We agree that a more comprehensive approach that would not need to distinguish between short and long memory (using a relatively arbitrary p-value, Figure 4) would probably be preferable, if it could be achieved with the same parsimony.

It would also be tempting to directly relate our CFC to distributions of travel time (see the discussion in section 1.2). But this is clearly out of the scope of our method: We do not follow any water particle from its entry to its exit as a tracer would do. Thus, the values of memory that we obtained may not reflect water ages.

## 4.3 Perspectives

In order to be used operationally, it would be interesting to predict catchment memory without the need for model calibration

against long time-series of discharge observation. An efficient regionalization of the approach could rely on defining relations between the characteristics of the CFC and hydroclimatic characteristics. This study shows that elasticity may be regionalized properly, as it is mainly driven by climatic conditions. However, the parameters of the Gamma distribution seem less easy to





regionalize, especially the scale parameter $\beta$ (see Appendix B). One perspective would be to better relate the two parameters of the Gamma distribution to catchment characteristics or to try other distribution assumptions that would allow for this more easily.

The parsimony of the proposed approach allows us to consider large-scale implementation on well-gauged territories. From a hydrological forecasting perspective, the maps thus produced can be used to better identify the predictability of the hydrological behavior of a catchment through the knowledge of past climatic inputs. From a changing climate perspective, they also provide an initial understanding of the sensitivity of watersheds to climatic anomalies and their effect on a multi-year time scale.

Catchment memory is fundamental for hydrological understanding, especially of regional hydrological processes. We need to learn more about this phenomenon for efficient water management and landscape planning. To better understand the causes of differences in catchment memory and elasticity, we recommend a more thorough analysis against physiographical conditions, including different catchment characteristics mentioned in section 1.3. This could, for instance, be geomorphology, geology, vegetation, soil type and depth, aquifer interactions, groundwater levels, and human modifications (tile drainage, land degradation, reservoirs).

*Data availability.* French climatic and hydrological data are provided by Météo-France and SCHAPI. French discharge time series are available from the French Hydro database (hydro.eaufrance.fr). Swedish climatic and hydrological time series are provided by the SMHI (https://www.smhi.se/data/hydrologi/vattenwebb).

## Appendix A: Relations between elasticity and catchment memory

One objective of this study is to disentangle catchment memory from catchment elasticity. This appendix illustrates that no strong relations were found between these two indexes, neither for catchments with nor for catchments without pluriannual memory.





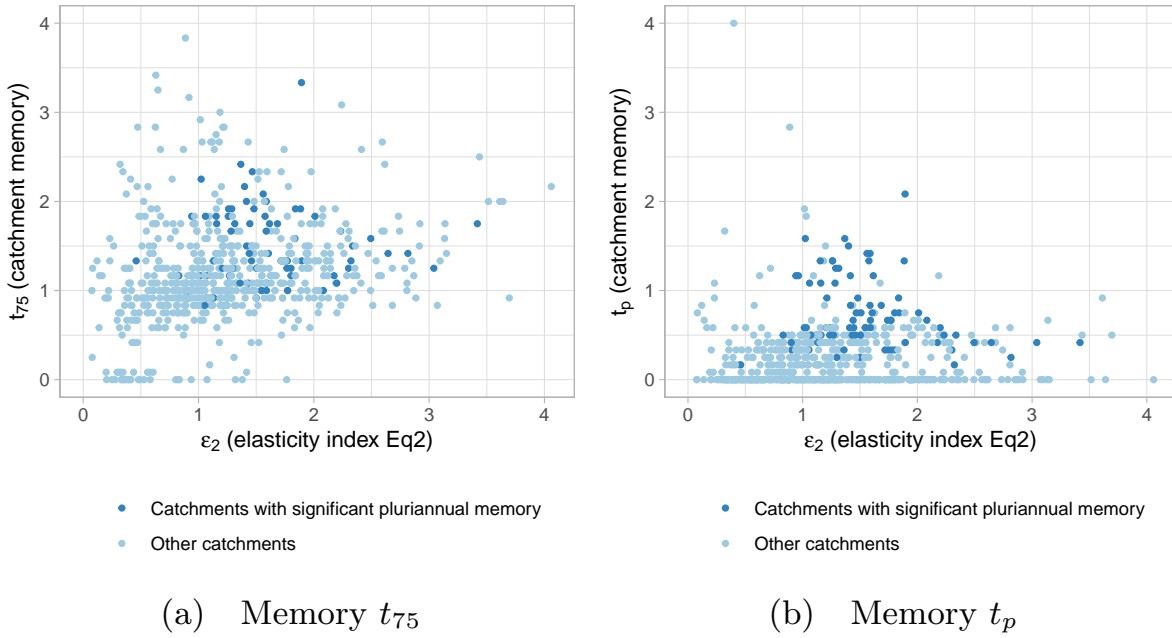

(a)   Memory $t_{75}$                   (b)   Memory $t_p$

**Figure A1.** Relation between elasticity index ($\varepsilon_2$) of equation 2 to catchment memory assessed by $t_{75}$ and $t_p$

**Appendix B:  Regionalization of Gamma distribution parameters**

In this work, we assumed that the CFC can be described by a Gamma distribution. This appendix provides the spatial distri-
bution of the two parameters of this Gamma distribution ($\alpha$ and $\beta$) optimized for each catchment (Figure B1). Figures B2 and
B3 illustrate the challenge of regionalizing these CFCs, with opposite relationships between France and Sweden when relating
these parameters to hydroclimatic characteristics.





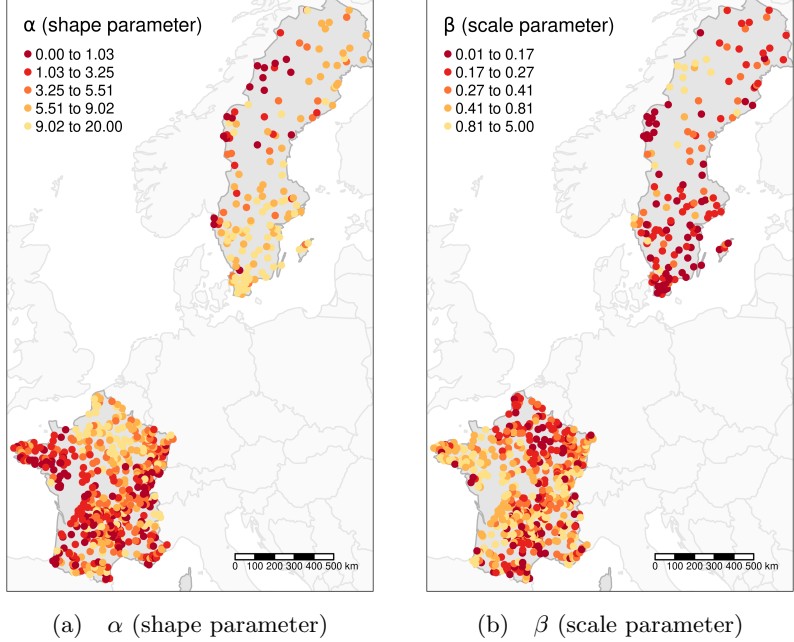

(a)  $\alpha$ (shape parameter)  (b)  $\beta$ (scale parameter)

**Figure B1.** Map of the parameters of the Gamma distribution (equation 3) calibrated inside equation 2. The classes are delimited by quantile values.





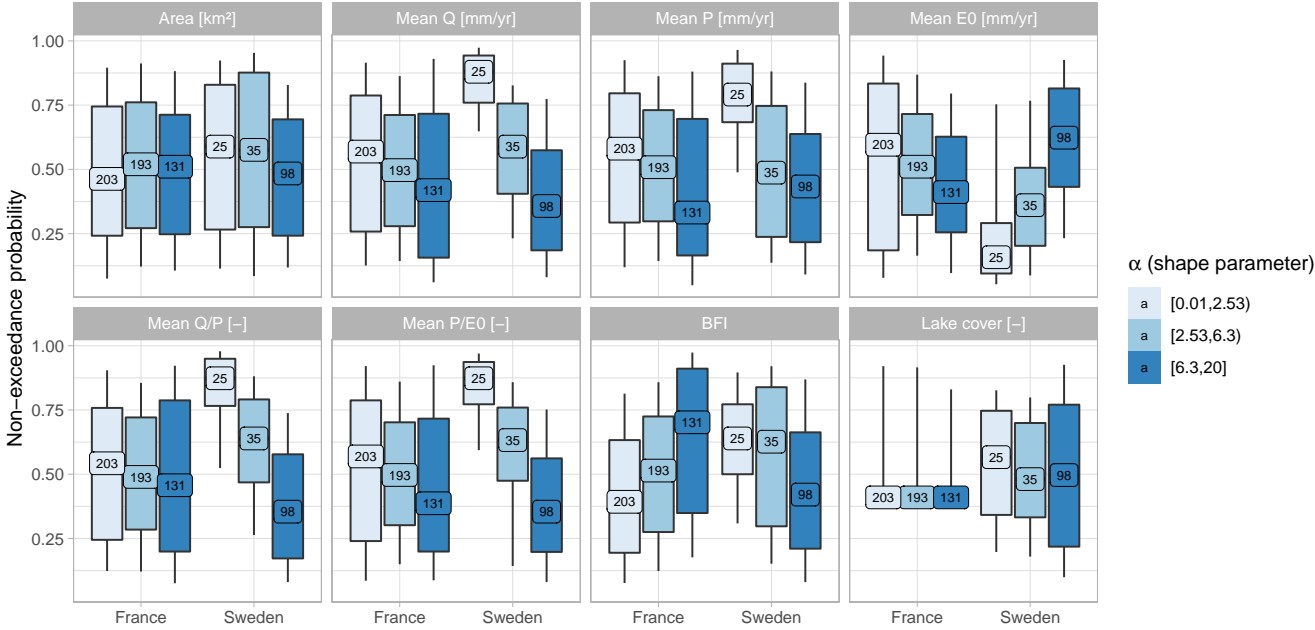

**Figure B2.** Distribution of hydroclimatic characteristics according to three classes of the shape parameter $\alpha$ of the Gamma distribution (equation 3) calibrated inside equation 2. The classes are delimited by quantile values. The numbers within each boxplot describe the number of catchments.



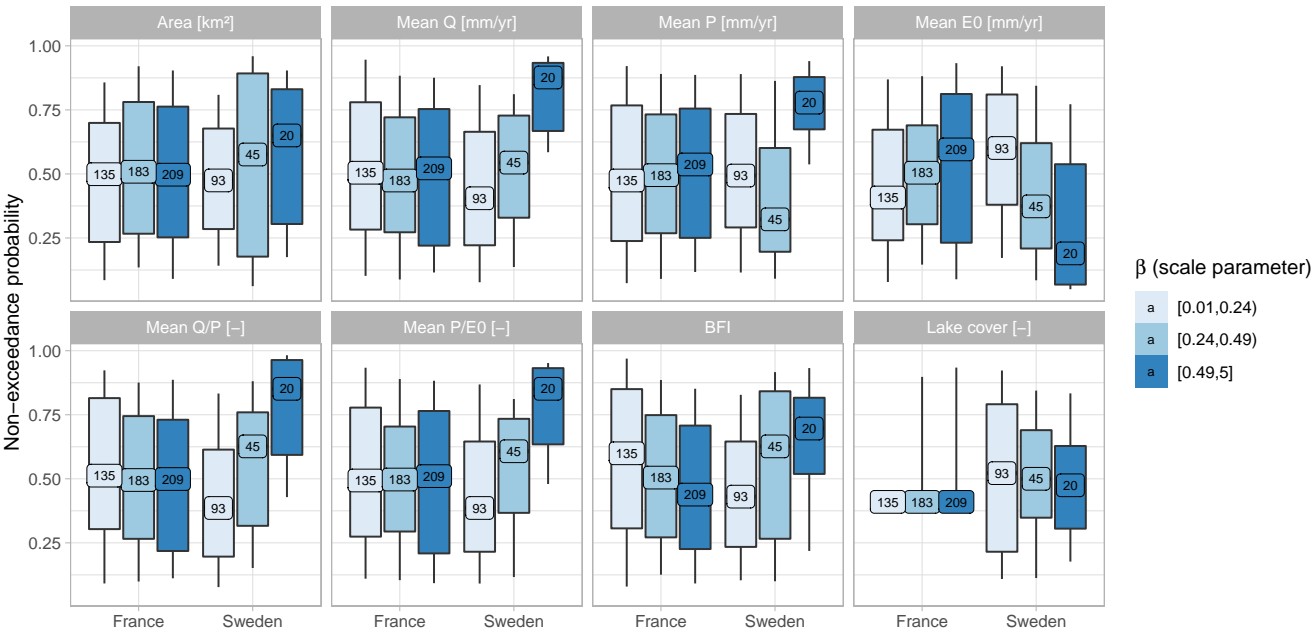

**Figure B3.** Distribution of hydroclimatic characteristics according to three classes of the scale parameter $\beta$ of the Gamma distribution (equation 3) calibrated inside equation 2. The classes are delimited by quantile values. The numbers within each boxplot describe the number of catchments.

*Author contributions.* AL, VA and LC designed the study and performed the analysis. GL and BA provided the Swedish data set and analysed the results in Sweden. AL wrote the paper, and all authors provided input on the paper for revision before submission.

*Competing interests.* The authors declare that they have no conflict of interest.

*Acknowledgements.* This work was funded by the project AQUACLEW, which is part of ERA4CS, an ERA-NET initiated by JPI Climate, and funded by FORMAS (SE), DLR (DE), BMWFW (AT), IFD (DK), MINECO (ES), ANR (FR) with co-funding by the European Commission [Grant 690462]. The authors gratefully acknowledge the reviews of Gaëlle Tallec, Charles Perrin, and Antoine Pelletier.





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
