# Peer review of "Quantifying multi-year hydrological memory with Catchment Forgetting Curves"

_Hydrology and Earth System Sciences, 2021_

## Author Response (AR1)

**Answer to reviewers**

**Quantifying pluriannual hydrological memory with Catchment Forgetting Curves**

Alban de Lavenne[1,2], Vazken Andréassian[2], Louise Crochemore[1,3], Göran Lindström[1], and Berit Arheimer[1]

[1]SMHI, Norrköping, Sweden
[2]Université Paris-Saclay, INRAE, UR HYCAR, Antony, France
[3]INRAE, UR RiverLy, Lyon, France

**Correspondence:** Alban de Lavenne (alban.delavenne@inrae.fr)

**Comments from handling editor**

> This paper received three detailed reviews and a community comment. The reviewers were rather critical about the scientific quality of the paper (judged as "fair" by all of them); all three reviewers recommended major revisions and I invite the authors thus to carefully revise the paper along their answers in the public discussion and to consider in addition my review comments below (adding to and going beyond the reviewers' comments). The paper will undergo re-review.
>
> The formal rebuttal should not just copy the public discussion but be much more precise about how the manuscript was revised, with clear indications about what was modified were (this is not always explicitly done in HESS; however, the public discussion is elusive at times regarding actual planned modifications and detailed indications (with line numbers) on modifications greatly help during the re-review. I thus strongly encourage to submit a detailed rebuttal with line numbers and a track-change version
>
> Below some additional comments that I would like to invite you to consider during the revision.

Thank you for handling this review and for your additional comments. We have addressed each comment one by one below in more detail than in the online discussion and described the changes to the manuscript.

> The revised version should state more clearly that you use hydrological years (it triggered questions, thus the vocabulary should be clearer throughout).

The paper now clearly states *Daily hydroclimatic data were aggregated at the annual scale for the purpose of this study, with the hydrological year starting on October $1^{st}$. By defining the start of the year in this way, rather than by a calendar year, we aim to minimise a water volume that could be carried over two calendar years.*

Furthermore, you might want to discuss whether the fixed-date hydro year (1 October) is a good choice for all studied catchments or whether it is a compromise at regional level. In Switzerland, we also use 1. It certainly the best choice for snow-dominated catchments since it mostly avoids carry-over effects of snow. For pluvial catchments, however, the summer drought can continue into early November or on the contrary be stopped by large precipitation amounts in autumn. Accordingly, the assumption of low interannual storage variation does often not hold. This means that choosing 1 October as start date of the hydrological year does not have the same effect for different hydrological regimes; for some, storage carry over is minimized for others not. This will impact the memory estimate.

10    The choice of a fixed date for the hydrological year is indeed a compromise that we have made for all catchments. We investigated the impact of this assumption on our analysis by using alternative definitions of the hydrological year (Figure 1 and 2): starting the hydrological year on 1 August, 1 September and 1 November. Changing this date will indeed slightly impact the analysis, but the agreement on the detection of multi-year memory is above 90% and the trends are very similar. In the light of these additional results, we think that using 1 October remains a reasonable assumption.

15    Moreover, as this article demonstrates, there are catchments where a volume of water is carried over several years, which in any case underlines the fact that a complete water balance cannot be achieved over a 365-day period.

The manuscript now states: *This start date is a compromise across our entire data set and a sensitivity analysis has shown little influence of choosing earlier or later start months. Moreover, as this study aims to emphasize, a water balance at the annual scale can seldom be comprehensive.*

[Figure]

**Figure 1.** Impact of the start date of the hydrological year on the RMSE of the simulated Q/P

[Figure]

**Figure 2.** Impact of the start date of the hydrological year on the average catchment memory for catchments with multi-year memory

> The above also goes into the direction of the comment of reviewer 2 on the fact that the annual time step might not be the ideal time step to answer the research question.

This time step is a deliberate and strategic choice to answer our question. Our research question is focused on multi-year memory so we deliberately aim to remove signals of seasonal memory from our time series. Although we agree it is tempting to work at a finer time step, such as a monthly time step, this would imply a more complex modelling. Explaining monthly anomalies would require a different model able to describe the intra-annual storage/release dynamics as well as this multi-year 25 water storage that we aim to qualify here. By using an annual time step, we aim to extract only the information we need from

the data and provide a parsimonious description of this long term memory, and complement existing descriptions of seasonal memory (e.g. Iliopoulou et al., 2019).

> You did not attempt to link the memory to snow storage. This seems like one of the drivers of memory: different relative amounts of winter snowfall might have a different recharge effect. Snow is only mentioned once in the paper.

Thank you for pointing this out. We were also expecting some spatial pattern where snow accumulation is important (Alps and North of Sweden). However, it appears to be mostly seasonal memory without a significant impact on Q/P over several years.

The manuscript now states: *No spatial pattern appears in snow-covered regions (e.g., the Alps and northern Sweden). This shows that snow melt affects mostly the seasonal memory with no significant impact on $Q/P$ over several years.*

> We need more details about the used Baseflow Index. How does the method deal with high baseflow during snowmelt periods? Is it meaningful for snow-dominated catchments?

The manuscript now states: *Finally, the contribution of groundwater is assessed by a baseflow index (BFI) calculated according to the work of Pelletier and Andréassian (2020) where the baseflow is estimated from the outflow of a quadratic reservoir. The approach has two parameters (calibrated for each catchment): The reservoir capacity and the time depth over which past effective precipitations filling this reservoir are taken into account. This baseflow filtering was performed with the associated "baseflow" R package (Pelletier et al., 2021).*

The methods should thus be meaningful even for snow-dominated catchments. Indeed, the parameter defining the time depth, over which past precipitations are taken into account, is calibrated on each catchment. This lag between snow accumulation and snow melt is thus considered.

> What enters the estimation of $E_O$? Not ideal if the reader has to look it up; if e.g. temperature dominates the estimate, this should be explicitly mentioned.

The manuscript now states: *For all stations, potential evaporation ($E_0$) was estimated from daily mean air temperature and daily extraterrestrial radiation following Oudin et al. (2005).*

How can a Gamma distribution be defined for negative values as is stated in answer to CC1? The Gamma distribution is defined for positive real numbers only. And: the Gamma distribution is used here to describe a functional relationship that is not a probability density function, this should perhaps be made clear somewhere. When it is used in the context of transit times, it describes the distribution of transit times. And I would avoid talking about a "skewed distribution", you use the Gamma distribution not as a distribution as far as I understand. This also means that it should not read "we fit a Gamma distribution" (which would imply e.g. maximum likelihood estimation) but that you optimize the parameters such as to reproduce the omegas; besides: what is meant by "We found that a simple exponential parameterization would not be enough as it does not allow lags." The lag is not explicitly modelled in the used functional relationship. It becomes clear later on; should read as "a simple exponential function would not allow to describe a maximum at time steps different from zero, i.e. not allow to describe a lag between $Q/P$ and $P/E_0$

Our answer to CC1 may have been confusing. CC1 asked why Gamma distribution is not 0 at year zero. In our answer we aimed to explain what appears in the figure 3 below: depending on the value of $\alpha$ and $\beta$, the Gamma density function could be above 0 for year 0, with the shape of an exponentially decreasing function. The aggregation time step also plays a role as suggested by CC1.

[Figure]

(a) Gamma density function                (b) Inferred values of $\omega$

**Figure 3.** Disentangling Gamma distribution and values of $\omega$

In the previous version of the manuscript, the different values of $\omega$ for each year $i$ were directly extracted from the Gamma density function with a lag of one year ($i + 1$): This lag allowed us to have more flexibility on the possible values of $\omega$ for year 0. However, the different comments led us to update this strategy: We now extract the different values of $\omega$ for each year $i$ by integrating the Gamma density function between $i$ and $i + 1$. This new strategy brings very similar results in terms of performance and $\omega$ values of the CFC (see Figure 4). However, based on the reviewers' comments, this strategy seems simpler and more appropriate for handling the Gamma density function. All figures in the manuscript have been updated

accordingly, but this creates only minor differences. Figure 3 shows an example of how the $\omega$ values are extracted from the Gamma distribution.

The manuscript now states: *The different values of $\omega$ for each year $i$ are estimated by integrating the Gamma density function* between $i$ and $i+1$. *These $\omega$ values are rescaled so that their sum is equal to 1, as defined in equation 2, and to provide the final values of the CFC. In summary, a CFC is built from the optimisation of equation 2 using three parameters ($\varepsilon_2$, $\alpha$ and $\beta$).*

[Figure]

**Figure 4.** Difference of RMSE on the simulated Q/P between the two ways of extracting $\omega$ values from the Gamma distribution.

Regarding the vocabulary that we use, we agree that saying "fitting a statistical distribution" might be confusing, as we do not have a population of memory estimates with associated frequency. We now rather say that we calibrate the two parameters of the Gamma distribution that allows us to derive the $\omega$ values obtained empirically. Moreover, we do not use the expression "statistical distribution" anymore to talk about the CFC, we now prefer expression "temporal distribution" to refer to the timing (lag from year 0), more or less like a unit hydrograph.

> I would avoid using the term "continuous" curves since you use a continuous function that is, however, defined only at discrete times (at time steps corresponding to entire years); what means actually 1.5 years (e.g. for the extraction of $t_{75}$) if the data time step is 1 year?

We now use the term "continuous" only when we do not use it at annual time step. This extraction of $t_{75}$ is an extraction of the Gamma distribution that can be generated after the calibration of $\alpha$ and $\gamma$ and which is defined for any memory value. We use it to extrapolate our description of catchment multi-year memory, as it allows us to extract any quantile value. The fact that it is an extrapolation is now better stated in the manuscript. We think that the shape of this Gamma distribution is still informative, despite the fact that the calibration has been performed at an annual time step, and allows us to give more nuance in the multi-year memory description.

The manuscript now states: *In order to quantify catchment memory, we assume that the Gamma distribution, from which $\omega$* values have been extracted, can be used to extrapolate a continuous temporal distribution of catchment memory.

> "we extract two characteristic times from the continuous distribution": it is simply a continuous function not a distribution, or are omegas interpreted as probability densities?

We do not interpret $\omega$ as probability densities. We are interested in the shape of this Gamma distribution that continuously describes the contributions of all previous years (our CFCs). Therefore, we do not seek to extract probabilities, but rather weights to built a temporal distribution (that we may also call "response time" distribution). However, we use the Gamma distribution to extrapolate what has been calibrated at an annual time step: In the same way that we extract $\omega_i$ by integrating the Gamma distribution between $i$ and $i+1$, we look at the moment when this integral reaches a given quantile value (75%, for $t_7 5$).

> Line 207: "For the sake of simplicity, before fitting a Gamma distribution, we first fit a simple annual elasticity model (a zero-memory model), and use a statistical significance test (Student's t-Test with pvalue < 0.01), to decide whether equation 2 improves significantly on equation 1." Can you explain what you actually test here, what is meant by "improve"? Model 2 has more parameters than model 1, so it necessarily improves. What values are compared with the t-test?

Thank you for pointing this missing information. We now have specified that our objective function is a root-mean-square error (RMSE) of the $Q/P$ anomalies. We use it for parameter optimisation as well as for model selection. The t-test allows us to compare the RMSE values obtained with the two models.

> The Math notation should be improved as mentioned by one of the reviewers. I recommend using more classical notations and use appropriate indices. This should ease some of the explanations (e.g. what means a "dated value").

We followed the recommendations proposed by the anonymous referee #3. We removed the expression "dated value", as we have now a time index $i$ in the equations.

**Review from anonymous referee #1**

The main result that elasticity values are underestimated using a normal approach seems questionable. Eq. 1 quantifies how $Q/P$ of year X varies with $P/E_o$ of year X. These numbers are both hopefully of a similar sign typically. If the same is quantified using Eq. 2, the $P/E_o$ values originate from multiple years, thereby having values that will often differ in sign from $Q/P$ of year X. Since their combined weight (that is $\sum(\omega_i) = 1$) is still 1, the associated elasticity value needs to be higher yield a similar effect of $P/E_o$ on $Q/P$. A difference between $\varepsilon_2$ and $\varepsilon_1$ would therefore have little to do with physics, but rather (partly) arises from a mathematical artifact. This seems supported by the fact that even in catchments with no significant memory the $\varepsilon_2$ still typically very strongly exceeds $\varepsilon_1$.

Thank you for pointing this out. This is an important point that the manuscript probably did not address in sufficient detail.

This is basically what Figure 6 shows, and we agree that $\varepsilon_2 > \varepsilon_1$ could be explained mathematically. Equations 1 and 2 both quantify how $Q/P$ varies with $P/E_o$. However, Eq. 1 uses only one $P/E_o$ value whereas Eq. 2 uses a weighted average of past $P/E_o$ values. We could expect this averaging to smooth the values of $P/E_o$ and, because we use the anomalies of these ratios, to lead to a value closer to zero (which is the long-term average value of $P/E_o$ anomalies). Lower absolute value of anomalies of $P/E_o$ could then be compensated by higher elasticity values.

Although it is mathematically logical, we do think that it is interesting to discuss about how we do evaluate this elasticity when catchments have a long-term memory. The important buffering role played by the catchment may prevent a correct estimation of the elasticity when considering each year independently (like in Eq. 1), which is the reason why we propose Eq. 2. We do not aim to emphasise the absolute difference between $\varepsilon_1$ and $\varepsilon_2$ as a main result of this paper (we may even avoid absolute direct comparison if it brings confusion), we rather want to emphasise the relative difference between these two approaches. This is done by the relative comparison of spatial distribution in Figure 9. Figure 6 also shows with two colours that catchments with multi-year memory usually have higher relative differences (in the sense of a distance to the abscissa) than the rest of the catchments. In this sense, we mean that the elasticity values may be underestimated when catchment memory is ignored.

We have updated the discussion on that point. The manuscript now states: *The numerical values of the elasticities obtained with equation 1 and 2 should not be compared (the fact that one is lower than the other has no meaning): Equation 1 uses annual $P/E_o$ anomalies whereas equation 2 uses a weighted average value of past $P/E_o$ anomalies. The averaging of past $P/E_o$ anomalies will inevitably smooth the extremes and will give a value generally closer to zero (which is the long-term average value of $P/E_o$ anomalies). These lower anomalies of $P/E_o$ are logically compensated by higher elasticity values during calibration ($\varepsilon_2 \geq \varepsilon_1$).*

*Figure 6 also shows that catchments with multi-year memory usually have higher relative differences between $\varepsilon_1$ and $\varepsilon_2$ (in the sense of distance to the bisector) than the rest of the catchments. This highlights that, despite the numerical artefact*

*previously discussed, elasticity of catchments with multi-year memory is often under-estimated if this memory is not explicitly considered. A climatic anomaly will thus affect runoff yield more strongly than expected by equation 1, but with a delay.*

> It is unclear how it is possible that so often a particular year's aridity explains that year's runoff ratio so poorly. Is this because the paper does not make use of water years, but calendar years instead?

This is not the case: as you did expect, a particular year's aridity explains usually very well that year's runoff ratio (also, we do use water years). Only a few catchments show a lack of relationship (the relationship is then lagged by one year), but it is the exception and not the rule. You may have got this impression because we used one of these exceptions as example in Figure 1. This is probably a bad choice and we added a "normal" example in the revised version: *Medstuguån* River (225 km$^2$) in Sweden where no multi-year memory is detected.

> The writing is at times unclear. I made many detailed comments below, but those are not necessarily comprehensive in resolving all issues.

We have included all the comments below. We hope that the writing is easier to understand now.

> Abstract: I think the relevance of the study would become a lot clearer by starting the abstract by introducing the problem that this paper addresses (e.g. a knowledge gap, or a paradigm that is challenged), rather than just stating what has been done.

The abstract now starts with: *A climatic anomaly can potentially affect the hydrological behaviour of a catchment for several years. This article presents a new approach to quantify this multi-year hydrological memory, using exclusively streamflow and climate data. Rather than providing a single value of catchment memory, we aim to describe how this memory fades over time.*
We think that the last sentence of the abstract is also relevant on that point: *Our work underlines the need to account for catchment memory in order to produce meaningful and geographically coherent elasticity indices.*

> L2: it is a "precipitation – runoff" relationship as many catchments will also experience snow.

We have replaced "rainfall–runoff relationship" with "precipitation–runoff relationship" throughout the manuscript.

> L2: For clarity: rather than saying "focusing on" just describe what elasticity actually expresses.

We added the definition of elasticity.

135

> L4: since "humidity" can refer to several hydrological conditions, I'd more accurately introduce this concept.

We need to keep the abstract short and therefore adjust the level of detail accordingly. The abstract has already been extended based on previous remarks. We think that the reader can refer to the manuscript to have our definition of humidity index and how it has been computed.

> L5: make "distribution" plural to indicate that each CFC has its own parameterization.

140  Thank you. We now used plural.

> L5: rather than say that a gamma distribution is used, provide some context of why a gamma distribution is used (e.g. it fits the data?).

The choice of a gamma distribution is a modelling assumption. It is indeed the most efficient distribution we have found. We discuss that in the methods. We prefer to keep the abstract as simple as possible and so we would prefer to avoid this level of detail in the abstract. Gamma distributions are commonly used in hydrology so we do not want to insist on that in the abstract.

145

> L7: what are: "powerful aquifers"?

We mean aquifers that strongly impact the hydrology. We replaced "powerful" by "large".

> L7: "a long memory" can be made more specific and thereby more informative.

We added *i.e. with the impact of climate anomalies detected over several years*.

> L8: state how aridity matters rather than that it matters.

150  We added *with drier regions exhibiting longer memory*.

L8: I am unsure what "appears to be one of the main drivers" really means here. Please rephrase it to be more accurate of how it matters.

We mean that, among the different catchment descriptors, aridity is the descriptor for which we found the strongest relationships with memory. We have rephrased it that way: *For both countries, a relatively strong relationship between the aridity index and memory is identified, with drier regions exhibiting longer memory.*

L8-9: "Our work underlines the need to account for catchment memory in order to produce meaningful and geographically coherent elasticity indices." Sounds like a nice conclusion but it does not seem to reflect that >80% of the catchments have no significant memory effect... This should be discussed in the abstract.

Thank you, this important information was indeed missing. We have added it in the abstract. It is true that the majority of catchments do not have a strong memory. However, even if this only concerns 15 % of the catchments, the elasticity indices were clearly implausible if memory is not taken into account. This percentage may also change with the data set. Nevertheless, we agree it is important to highlight this result in the abstract and now do so.

L15-16: I think a reference or two would not be inappropriate here.

We now cite Andréassian and Perrin (2012) where, within the Turc-Budyko framework, $Q/P$ is related to humidity indices $P/E0$.

L20: "will" seems redundant.

We have suppressed it.

L21-24: I find it hard to fathom the statement "To make this discussion of a complex matter simple, we start with a first-order simplifying assumption: We hypothesize that a catchment may have both a short-term and a long-term memory (see e.g., Risbey and Entekhabi, 1996; McDonnell, 2017); we consider the short-term memory to be seasonal, and will not address it in this paper in order to focus on the long-term (pluriannual) memory effects.". To me, this statement is unclear (how are seasonal and longer-term memory really separate?), it is not clear why the assumption you make can be made (because it is not explained), and the reference seems off (why refer to a paper about water ages, when the quantity you're interested in are quantities of water?).

We agree that it would have been better to have a model that would have provided a comprehensive description of catchment memory, from seasonal to long-term memory. This is why we clearly point out this limitation here, as well as in the discussion with the citation of Klemeš et al. (1981). This sentence you quote mainly aims to avoid confusion about what the reader should expect about the level of detail that we are addressing concerning catchment memory. It is not a starting assumption that should be accepted or rejected before going further. We proposed an approach that works at annual time step and we emphasize that seasonal memory is consequently not described. The temporal discretisation will only allow detecting what we call "long-term memory". We invite further studies that could also quantify seasonal memory in the perspective of this work.

The work of Risbey is one example that shows the possibility to study inter-annual variability of annual streamflow from the variability of annual precipitation. By quoting McDonnell, we want to emphasise that there is a long-term dynamic (over several years) that is not taken into account when the water balance is made by considering each year independently. This idea reflects our philosophy that elasticity analysis should also take into account these long-term dynamics.

We have reformulated the sentence that way: *To make this discussion of a complex matter simple, we distinguish short-term memory from long-term memory. We consider the short-term memory to be seasonal, and because we decided to work on an annual time step, this will not be addressed in this paper. We focus on the long-term (multi-year) memory effects, where instead of analysing each year independently (see e.g., Risbey and Entekhabi, 1996), we aim to take into account previous years to better explain inter-annual variability (following McDonnell, 2017).*

L26: "its variability" in space, in time, or both? Please specify.

We specify "spatial variability" (among catchments).

> L28-29: I find it hard agree with the statement "is obviously a function of catchment storage capacity (in groundwater aquifers, wetlands, lakes or glaciers)". It is not the "capacity" that matters, but rather the "storage amounts" which are largely independent of "capacity". For example, there is a lot of storage capacity in the pores of Sahara sand, but only if these pores are filled (or not) will influence whether it has an influence on memory.

Yes thank you, we agree on that. We just use "catchment storage" now.

> L29-30: "the originality of this paper will be in the quantification of forgetting curves at catchment scale:". I understand that this concept is original, but I think it needs to have more context of why this concept is useful compared to current knowledge. The latter is lacking from this part of the introduction, and only is introduced later. Putting this upfront will help the reader not being confused why this study is undertaken at all.

It's true that at this point of the paper, the originality is not fully understandable. We now briefly mention that current knowledge usually only provides memory indices or single value (referring to section 1.3), whereas this work aims to describe memory dynamics over time.

> Section 1.2. I think this clarification does not need an entire section, but should be resolved in a single sentence (or maybe two at most). Once this is resolved, I would recommend to also remove any travel time stuff from the following section(s) as this is a separate topic that is not addressed in this paper.

We think it's a critical distinction that needs to be made to avoid any confusions of what we have achieved (and not achieved) in this work. We chose to create a separate section because we think that "water age" and "catchment memory" are often mixed up (in a similar way to velocity and celerity). For this reason we prefer to keep a full subsection for this. The online discussion with Tomas Over is a good example of that.

> Section 1.3: The statement that "existing methods aiming to analyze memory either summarize the memory by a single value and/or provide an index that cannot be directly interpreted as duration" provides (in theory) a clear motivation for your study. If you also state this at the start of section 1.3 the reader will much better understand what is lacking in these pasts works (rather than concluding it in hindsight). In general, this section can be condensed.

We agree, it's a key point so we try to better emphasize it now. We added this in the abstract and at the end of section 1.1. Note that it is also mentioned in the first goal of this paper (section 1.6).

> Section 1.4 and 1.5: this a description of why people have reported catchment memory before, but I am unsure how this is useful (in this format) as the introduction of the paper. Can it be reframed to introduce your work, rather than mostly just listing findings? Also, I do not thing that listing flood effects or water quality affects is useful here as these topics are not addressed in your own work.
>
> I understand that the above suggestions may sound a bit arbitrary, but I think you'd do the reader (and therefore your own paper) a huge favor by having a more to the point introduction.

We agree that our review of the literature is rather broad. We think it is interesting to have this overview of the subject, as this topic is addressed in many different ways. In order to avoid losing the reader, we have structured this introduction into subsections.

Section 1.4 aims to provide an overview of why the topic of "catchment memory" is worth studying. Mentioning that this topic is useful for predictability of discharges or legacy of pollution offers different perspectives and highlights that this topic is being addressed by different communities.

Section 1.5 is a section about the drivers of catchment memory. We do think it is interesting to keep it, as it leads us to the choice of some catchments descriptors that seems relevant to have in our analysis. It provides insight for our analysis on the effect of catchment size for instance, or humidity index for instance.

For these reasons, we think it would be a shame to remove these sections from the paper. However, some references have been removed to make it more concise.

> Section 1.6 I think the study area needs to briefly mentioned with the scope, as this defines the scope of the paper.

For the third goal of this study, which aims "To provide some physical indicators of the main drivers of memory and elasticity", we added that this will concern only France and Sweden. The first and the second goal does not depend on the data set.

> L158: "that are not regulated" how is this defined?

It is a classification provided by the databases we have used. We now added *The degree of regulation is the percentage of the volume of the mean annual flow that can be stored in reservoirs located upstream of the gauged catchment outlet.*

> L167: "We accepted a maximum of 10% of missing data per year". OK, but what did you do with these missing data? Just calculate annual Q over fewer days?

It is averaged over fewer days and it is rescaled to provide an estimate over 365 days. We added this information.

> L167: "respect"?

We changed "respect" by "observe".

> L168: "in order to be able to" or simply "to".

Yes thank you.

> L181: are these calculated over calendar years of based on hydrological years? The latter seems more useful?

It is computed over hydrological years starting on October $1^{st}$. This information is already provided before, in the third paragraph of section 2.1. See also the discussion above with the co-editor about the sensitivity of the choice of the starting date.

The manuscript now states: *Daily hydroclimatic data were aggregated at the annual scale for the purpose of this study, with the hydrological year starting on October $1^{st}$. By defining the start of the year in this way, rather than by a calendar year, we aim to minimise a water volume that could be carried over two calendar years.*

> Figure 1: Can a more typical example be shown? A catchment that does not respond to its current year conditions seems like a (very strong) outlier?

Figure 1 now has two catchments: one Swedish catchment without long-term memory and one French catchment with long-term memory.

> Line 199: "we hypothesized (after many attempts that we cannot report here)". I have no idea what has been done here, but to still call it a hypothesis seems like a stretch?

Yes we agree, it's not a naive hypothesis. Before proposing the gamma distribution, we tried exponential distributions (with and without lag), Rayleigh distributions and different other uses of the gamma distributions. We also tried to fit simple linear regression sequentially for each year but this lead to a much larger number of parameters (one per year). Some of these approaches can sometimes give results that are similar or better than the Gamma distribution for certain catchments. The Gamma distribution may also seem restrictive (for example, it does not allow negative values). For this reason we think that the choice of a distribution is always an assumption.

> L200: why mention transit time distributions? Transit times distributions have nothing to do with the presented study or approach so I am unsure why mentioning them helps?

We think that the difference between CFC and transit time distribution is tenuous. In this paper, we also wish to provide the reader with some thoughts about this distinction. Thus, despite the distinction made in section 1.2, we find it interesting to note that both descriptions can use the same distribution. Reviewer's comments below (#2) also seem to agree with the importance of this distinction.

> L203: What does "would not be enough" mean is this context. Please rephrase.

We now say: "*We found that a simple exponential parameterization would not be flexible enough as it does not allow lags (unlike a Gamma distribution).*"

> L216: Since there is no real reason for 75%? If you take 50% you can simply multiply alpha by beta, and choose a more typical percentage? Or, would it be possible to present a scatter plot of different percentages so it can be seen if this metric is robust?

The choice of 75% is indeed arbitrary, any other percentage can be extracted depending on the objective. We have tried several but we pick the third quartile as a fairly common value. We do not intend to claim that 75% is better than any other quantile and therefore justify it with additional graphics. This is mainly for illustrative purposes. The most comprehensive description of the memory is described by the CFC, but they can hardly be plotted on a map.

> L221: "This shows that pluriannual catchment memory is neither common nor very uncommon." Does it? Or does it suggest it's uncommon?

We now say: *This shows that multi-year catchment memory is not the most common situation.*

L244: "If larger catchments tend to have larger memory in France, this trend is not confirmed in Sweden" is unclearly formulated.

255 We now say: *Larger catchments tend to have longer memory in France, whereas in Sweden the memory does not seem to be related to catchment area.*

L245: earlier "humidity" was used and now "aridity"; please be consistent.

We have removed the word "aridity" throughout the manuscript to always prefer the word "humidity".

L246: "whereas the hydrological behavior under less humid climate is more variable and linked to the dynamics of long-term water storage" such an explanation might be feasible but there is no evidence supporting it. It is unclear to me whether this statement is considered a finding or speculation?

260 It is a proposal for results interpretation. The full sentence aims to drive the reader in that direction. We now say a bit more carefully: *It thus appears that the hydrological behaviour of the driest catchments is more dependent on past climatic conditions than that of humid catchments. It can be hypothesized that in wetter conditions, water storage is renewed more often and the memory therefore tends to decrease.*

L249: "clearly identifies" use a different verb (e.g. "is associated with")

265 Thank you, we have updated it with your suggestion.

Section 3.4. I do not think this a physically meaningful comparison. Eq. 1 quantifies how $Q/P$ of year X varies with $P/E_o$ of year X. These numbers are both hopefully of a similar sign. If the same is quantified with Eq. 2, the $P/E_o$ values come over multiple years, thereby having values that will often differ in sign from $Q/P$ of year X. Since their combined weight (that is $\omega_i$) is still 1, the associated elasticity value needs to be higher to still yield a similar effect of $P/E_o$ on $Q/P$. This seems like is has nothing to do with physics, but rather arises from a mathematical artifact. This artifact seems supported by the fact that even in catchment with no signifivant memory the $\varepsilon_2$ still always exceeds $\varepsilon_1$. Maybe I get it wrong, but please convince me so in a clear manner.

This was one of your major comment at the beginning of your review and we answered to it above.

> Figure 9: if these two values are compared, please show them on a similar color scale. However, as stated earlier, I do not think they are comparable.

As in the previous comment, we do not aim to compare absolute values of the two elasticities. As you noticed, they do not vary within the same range and may not be directly compared because of a numerical artefact. We rather want to emphasize a relative comparison of spatial patterns, which is the reason why we prefer to use different color scales (but with classes always delimited with quantile values as mentioned in the caption).

> All appendices can be Supp Info?

We'd rather keep this information directly after the manuscript than in separate files, since it directly supports parts of the main text. We would therefore prefer to keep them as is.

**Review from anonymous referee #2**

> This study works on catchment memory and performs a kind of sensitivity analysis to quantify the influence of past conditions on streamflow variability. The general topic of the paper is in the scope of HESS. Catchment Forgetting Curves (CFC) are introduced as a metric to characterize catchments' memory. As other reviewers mentioned already some concerns, I try to focus on other details here. Overall, language and structure is a little bit cluttered, however I can follow the story of the paper, but some analyses should be revised. May be this is a personal issue, but I dont think that the word "pluriannual" is the best choice as "multi year" is more common in the community.

First of all, thank you for your time and all your suggestions to improve our manuscript.

"multi-year" is effectively more common in the literature, we just found "pluriannual memory" in comparison to "annual memory" easier to read. But we want to describe the same temporal aspect. "Multi-year" is now used through out the manuscript.

**Major comments:**

> L12-18: What about human water use? This aspect is also missing in the introduction (long term effects) (L103-110). I see the short paragraph (L139-144) about human influences on catchment memory, but this should be more integrated into the introduction (looks like a marginal note here). The word 'human' is only mentioned two times in the manuscript, I recommend to put more focus on this potential driver of catchment memory (at least in Introduction/Discussion).

We have tried to avoid dealing with human influences in order to focus (as much as possible) on the natural behaviour of the catchments. In order to qualify these influences, we refer to the two hydrological databases we used for this study (in France and Sweden) where some indicators are provided. The degree of regulation is the percentage of the volume of the mean annual flow that can be stored in reservoirs located upstream of the gauged catchment outlet. We agree that this type of indicator does not fully qualify all possible human influences on hydrology and we share this view on the importance of this topic. However, this issue is beyond the scope of this paper where we specifically try to exclude influenced catchments.

The manuscript now states: *The degree of regulation is the percentage of the volume of the mean annual flow that can be stored in reservoirs located upstream of the gauged catchment outlet.*

> L36-50: The difference between water age and catchment memory is very important to explain. Authors can consider to embed research in this field in other studies, e.g. different storage concepts in Staudinger et al.(2017). Catchment water storage variation with elevation. Hydrological Processes, 31(11), 2000-2015.

Thank you for supporting this distinction between memory and water age, we also think it is crucial to understand this work. Thank you also for suggesting this paper, the different storage concepts provide a good framework of understanding. We added this aspect in the introduction.

The manuscript now states: *This distinction may also be linked to the different perceptual storages of water in a catchment. "Mobile storage", which controls transport in a catchment is more linked to water age, whereas "dynamic storage" which controls streamflow dynamics is more in line with our definition of catchment memory (see e.g. Staudinger et al., 2017).*

> L115-L119: I am not convinced here as there are a lot of studies finding large(r) groundwater storages in (relatively small) alpine headwater catchments (e.g., Staudinger et al., 2017 or other studies in Switzerland). Has the Merz et al. (2016) study in L120 multiple catchments in their analyses (with variation in size and elevation)? If so, is there a correlation between storages and elevation or area?

In our work, we were not able to demonstrate an effect of elevation on multi-year memory. The effect of snow accumulation is not even visible in Sweden where northern catchments (where snow accumulation is important) do not show longer memory than southern ones. If snow obviously impacts seasonal memory, it does not appear as a driver of multi-year catchment memory. However, our analysis looks at each catchment descriptor one at a time enabling only first order relation.

The manuscript now states: *Staudinger et al. (2017) found the largest dynamic and mobile storage estimates in high-elevation catchments.*

> L127-134: I am not sure if the examples from the Tropics and Sahara Desert are a valid justification of "baseflow importance".

We have removed this from the introduction.

> For me it looks like that the choice of "1 year" as temporal resolution may be not appropriate to answer the research questions: The "1 year" includes all effects 7 up to 17 months, "2 years" embeds everything from 18-30 month, right? This classification might be really critical and as the data allows for a more comprehensive analysis (e.g., seasons, months).

The elasticity analysis is performed at annual time steps using hydrological years. Thus, "year 1" includes all effects from October 1 (year i) to September 30 (year i-1); "year 2" includes all effects from October 1 (year i-1) to September 30 (year i-2). We believe that this is a relevant time step to study multi-year memory. We do not use a moving average strategy. We have deliberately chosen not to use a finer time step, as we generally follow the idea of the common annual water balance analysis where precipitation volume, precipitation and evaporation are assumed to be comparable. Seasonal storage dynamics are therefore not addressed by this study. It is discussed as a limitation/perspective of this work, as we believe that the proposed methodology could not deal with both memories at the same time.

This aspect of the definition of the hydrological year and the choice of the temporal resolution were also raised by the co-editor. We provided an answer and further analysis to address the sensitivity to our definition of the hydrological year in the answer above.

> The authors stated that short-term memory is not considered (L23+24) but in Fig. 5 I found a short- vs. long-term analysis. This is confusing. By the way where is short- vs. long-term memory defined?

In Figure 5, for illustrative purposes, we have divided the catchments with significant memory into two subgroups of equal size (short memory and long memory) as described in the caption. In doing so, we have sought to refine the analysis of

catchments with multi-year memory. However, the vocabulary in the introduction to the document can be confusing. In figure 5, we now use "short multi-year memory" and "long multi-year memory".

> It is stated that in Sweden 5% of catchments are regulated and that there is no regulation in French catchments (i.e., those are excluded). What kind of regulation is this and has it influence on the outcomes of the study?

325 We addressed this question also raised by CC1 above. We have considered the influence based on indicators provided by each database which are built on the influence of dams. We did not consider regulated catchments in order to focus on catchment natural behaviour.

> L245-248: This is not clear to me. Is about those drier catchments have a longer memory? If so, why they are drier as longer memory most likely come along with larger storages (which in turn will lead to more continuous flow, or?) Here more explanation is needed.

The analysis shows that in dry conditions, catchment memory is longer. We do not think that it means that water storage is
330 larger in dry conditions. We rather mean that in dry catchments, the hydrological behaviour is more impacted by past meteorological conditions, just because in humid catchments the wet conditions erase the impact of past conditions by "resetting" the storage states.

The manuscript now states: *For both countries, the memory increases in drier hydro-climatic conditions (as characterized by either lower discharge and precipitation, lower $Q/P$ or lower $P/E_0$). However, the effect of potential evaporation does not*
335 *appear clearly. It thus appears that the hydrological behaviour of the driest catchments is more dependent on past climatic conditions than that of humid catchments. It can be hypothesized that in wetter conditions, water storage is renewed more often and the memory therefore tends to decrease.*

**Minor comments:**

> L12: "biota", do you mean vegetation?

340 We have replaced "biota" by "vegetation".

> L19: 'past climatic sequences', could you please make a more precise statement about this?

We now used "past climatic anomaly".

> L24: Just a comment, the word 'pluriannual' is not very common, perhaps considering to switch to multi-year (cf. L88)

We have switched to "multi-year" throughout the manuscript.

> L162: What is exactly meant with 'not regulated' (only no dams?).

We addressed this question in your major comments.

> L165: Please state shortly the relevant variables to estimate E0.

We now say: *For all stations, potential evaporation ($E_0$) was estimated from daily mean air temperature and latitude following Oudin et al. (2005).*

350

> L172: How many Swedish catchments have what amount of lake area?

The median value of lake cover is close to 0.5 for Swedish catchments. In this section we do not aim to provide statistics of catchments descriptors, they are presented later in the figures 5 and 7.

> L184-190: How is the maximum of parameter w (=5) justified? I can think about some catchments that have 'a longer memory' than five years.

We found that catchments memory in our data set rarely exceeds 2 or 3 years. By setting the value of 5 years, we cover
355 the main range of possible memory values. Increasing this value would reduce the number of points that can be used in our elasticity analysis. For example, if you have a 25-years time series, you may have 20 years where the previous 5 years are available, or only 15 years where the previous 10 years are available.

We agree that even in our data set we have catchments where the streamflow is partly composed of water that is well beyond 5 years old. However, in the definition of catchment memory we have used, we were not able to detect an impact of annual
360 runoff yield.

> L192ff: Might be easier to understand to name it x- and y-axis although the description of axes is correct.

Without mentioning x- and y-axis, we now simply say: *the memory effect can be visualized by a series of plots showing the runoff yield anomaly as a function of the climate anomaly of the preceding years.*

> L221: This sentence is not clear to me: "This shows that. . . ."

365 We now say: *This shows that multi-year catchment memory is not the most common situation.*

> L254: "thinner soils"; is there data/analysis on that (in more detail)?

We now cite Ballabio et al. (2016) to refer to this aspect that we discuss qualitatively here.

> L285: "is spread out", perhaps consider to rephrase here.

We now say: *Even though these catchments have a longer memory of climatic anomalies, the impact of these anomalies is* 370 *distributed and smoothed over the years.*

> L326/327: Just a comment, perhaps a more in-depth differntiation between dry and wet years/seasons would be beneficial to better understand how variability in CFCs could be explained?

Variability of CFCs can only be explained by comparing catchments but not by comparing wet and dry years, as we do not formulate a CFC that can change over time. "Wet" and "dry" years are directly related to the anomaly: a positive anomaly is a wet year, a negative is a dry year.

375 We added this sentence in the perspectives: *We have proposed a static description of the CFC, future work could also examine how climate anomalies might alter the shape of the CFC over time.*

> Fig. 5: Might be helpful to switch to another graph type here as boxplots may hide bi-modal distributions. Perhaps violin plots are more helpful here or the data points can be added with a jitter to the visualization.

You will find a violin plot below for figure 5. This figure 5 already condenses a lot of information, we think that adding density distribution makes it difficult to read. We do not find clear bi-modal distribution that would justify that. We think that 380 quantile values are easier to compare, as proposed with the original visualization so we prefer to stick to it.

[Figure]

**Figure 5.** Distribution of hydroclimatic characteristics according to three classes of memory (described by $t_{75}$).

> Comments on the maps: I like the way French and Swedish catchments are compared with the point-maps. However, I suggest to reduce the point size a little bit to avoid too much overplotting. As the rest of Europe is not relevant for this study it might be also an improvement to have outlines of both countries next to each other to gain more space for the actual visualization (i.e., variability across the countries).

Thank you for this advice. Indeed, this issue of point size is a compromise between point visibility and potential overlap. We have reduced the point size on all maps.

We prefer to keep this European vision of the memory, the gain of space of having the two countries next to each other is done at the expense of the spatial continuity that is also interesting to keep, especially with regard to climatic descriptors. Both representations have their advantages and drawbacks, and we tried to find a compromise.

> Please revise paragraph structure (e.g., often line breaks seem to be redundant, for example in the abstract?)
> Examples: L156, L109

We have written two paragraphs in the abstract, one on the general approach and one on the results on our specific dataset. This was done deliberately. We reviewed the rest of the paper on this aspect.

 **Review from anonymous referee #3**

> This paper with the issue of catchment memory, it asks a novel question and is well in the scope of the journal. I have some issues that worry me, and should be clarified.

Thank you for your time and this feedback. We have addressed each point you have raised below.

> 1. I think the analysis should be done using hydrological years and not calendar years. It is common in hydrological modelling studies to refer to the years starting in September, where the catchment storage is low and so catchment memory. Using calendar years, there is a higher chance that the meteorological conditions from the end of the summer, when the wet season starts, will have an impact on the runoff in the following months. This effect is largely related to meteorology, and has little to do with the catchment memory that the authors are trying to investigate, which instead, should reflect a catchment property.

We fully agree that the analysis should be done using the hydrological year and not the calendar year, which is why we
395 start each year on October 1st. As answered to the co-editor and to reviewer #2, our study is based on an annual water balance analysis, which makes more sense if the hydrological year is used. It avoids issues such as snow accumulating in a year and only melting in the next.

This aspect of the definition of the hydrological year and the choice of the temporal resolution was also raised by the co-editor. We provided an answer and further analysis to address the sensitivity to our definition of the hydrological year in the
400 answer above.

> 2. Care should be taken to the fact that there is a spurious correlation between the variables Y=Q/P and H=P/E0, when such quantities are calculated for the same year. Both equations in fact contain the same variable P. I think the authors should recognize this fact and reflect on it, as it can have a strong impact on the analysis.

There is indeed the variable P in both sides of the elastic relationship between Q/P $\sim$ P/E. However, we believe that it has a limited impact on our analysis for two reasons:

- It is the same P on both sides only for the current year 0: it is Q/P of year 0 that is explained by the different P/E of the
405  previous years, which leads to a different P for years n-1, n-2... Moreover, it is reasonable to assume that each year is independent of the previous one (see comments on auto-correlation below).

- The analysis of the relationships of two correlated variables is not really a problem. We could have removed the P from one side and analysed Q $\sim$ P/E. This would have led to even more highly correlated relationships with respect to Q/P $\sim$

P/E. We chose to work with these ratios in order to study the relationship of two dimensionless variables. However, it is true that if Q and E were almost constant, the elastic relationship would look like f(x)=1/x and the elasticity index would be negative. Our analysis shows that this does not happen, and when catchment memory is taken into account, negative elasticity indices are no longer observed.

The manuscript now states: *As we chose to work with dimensionless variables $Y$ and $H$, one should notice that $P$ appears on both sides on the equation 2. However, they do have different time index $i$.*

3. An improved mathematical notation could help. Since the authors are working in two dimension, difference from average and difference in time, it would be helpful to explicitly write what e.g. delta is differentiating. Moreover, the capital delta symbol should be used, as this is standard when calculating discrete differences, perhaps with some subscript to indicate in which dimension the difference is calculated.

We thank you for this advice. We have updated the delta symbol and added a subscript in Equation 1 to better describe the time index.

4. Figure 1 shows that not only there is a lag 1 correlation between the Q-P and the P-E anomaly, but also that there is an autocorrelation of the P-E anomaly. This is largely an autocorrelation in climate properties, thus reflective of climate memory, rather than catchment memory. Such autocorrelation of climate should be analysed, and its effect removed or at least studied and recognized, otherwise there is a confounding effect.

We agree that it is necessary to check this autocorrelation of the climate inputs, in order to avoid the analysis of the climate memory instead of the catchment memory. We have carried out the analysis of the autocorrelation of the P/E and have not detected any significant autocorrelation (see figures below). The median value of the Pearson correlation between a P/E and the previous P/E is 0.05. For 90% of the catchments, this correlation is less than 0.2 and a statistical test shows no significant correlation, except for the few catchment areas in south-eastern France where the correlations are still quite low (and where no multi-year memory is detected). We can therefore reasonably assume that the memory we have detected is the memory of the catchment and not the memory of the climate. We now refer to this autocorrelation analysis by saying : *In order to avoid the detection of a memory in the signal $Y$ that would have been contained in the climatic input signal $H$, we checked that no highly significant auto-correlation was found in $H$.*

[Figure]

[Figure]

[Figure]

(a) Pearson's correlation coefficient with a one-year lag

(b) p-value of a correlation test based on Pearson's product moment

**Figure 6.** Auto-correlation analysis of annual P/E

5. It is unclear how the data are organized in order to enable the calibration of Equation 3. Moreover, it is unclear how epsilon2 is calculated. Finally, once the authors will explain how they sort the data into an histogram in order to enable the calibration of Equation 3, it is unclear why the omegas cannot be calculated directly from the histograms, thus without having to fit a distribution.

Equation 3 is calibrated along with equation 2, as equation 3 provides the different weights $w_i$ for each of the last 5 years (i = 0, ... n). The calibration is done by adjusting 3 parameters at the same time ($\varepsilon_2$, $\alpha$ and $\beta$) and by minimising the RMSE of the root mean square error (RMSE) of the Q/P anomalies.

We prefer to calibrate a Gamma distribution rather than calibrate each year independently as this reduces the number of parameters: the Gamma distribution has only two parameters whereas estimating the weight of each of the previous 5 years would require 6 parameters. It also provides a more consistent description of catchment memory.

See also our answer to the co-editor on that point. The objective function, the calibration algorithm, the extraction of the histogram from the Gamma distribution are now better described in the manuscript. In particular, the manuscript now states: *The different values of $\omega$ for each year $i$ are estimated by integrating the Gamma density function between $i$ and $i+1$. These $\omega$ values are rescaled so that their sum is equal to 1, according to equation 2, and to provide the final values of the CFC. In summary, a CFC is built from the optimisation of equation 2 using three parameters ($\varepsilon_2$, $\alpha$ and $\beta$).*

> 6. The results and discussion section poses several questions and corresponding analyses that are not anticipated in the method. Thus, that section reads more like a newspaper than like a scientific article. The question and analyses are interesting, but the methods should be organized to anticipate the structure of the analyses.

In the section "Scope of the paper", we wrote three objectives:

1. *To present a method, based on the concept of elasticity, that not only can provide an index relevant to catchment memory, but can also characterize its dynamic in a manner analogous to a forgetting curve (Ebbinghaus, 1885);*

2. *To disentangle catchment memory and catchment elasticity;*

3. *To provide some physical indicators of the main drivers of memory and elasticity for France and Sweden.*

We think that each objective is addressed with a proper methodology:

1. By proposing a formulation of CFC

2. By proposing an equation where $\varepsilon_2$ is distinguished from $\omega$ and by analysing each one separately

3. By looking at the relation between some catchments descriptors with memory and elasticity index

Indeed, the results and discussion section poses several questions that may seem like a less conventional presentation of results. However, they are all fully related to these general objectives:

- *Is pluriannual memory a rare phenomenon?* refers to objective 1

- *Where do catchments exhibit a pluriannual memory?* refers to objective 1

- *Can pluriannual memory be explained by hydroclimatic descriptors?* refers to objective 3

- *What do we miss in catchment elasticity analysis when not accounting for pluriannual memory?* refers to objective 2

- *Can elasticity values be explained by hydroclimatic descriptors?* refers to objective 3

We believe this structure makes the analysis easier and more pleasant to read, without compromising the scientific analysis we have conducted.

> 7. I am not sure that Figure 5 in the way it is formatted really conveys the message. Why not doing simply scatter plots, perhaps showing Spearman correlation values? I have the impression that this more classical way of plotting results might be more informative.

460    We tried different plotting strategies before proposing this one. Scatter plots are indeed the most direct way to approach the relationship between two variables. However, this visualisation is sensitive to outliers and the scatter of the 685 points makes the general trend difficult to visualise. We have therefore chosen to summarise the distribution by a boxplot, making the relationship much easier to interpret in our opinion. It also allows showing more information in one graphics, which is not possible with scatter plots. The scatter plots of $t75$ with catchment descriptors are provided below.

[Figure]

**Figure 7.** Scatter plots of the memory quantile $t75$ with the different catchment descriptors

**References**

Andréassian, V. and Perrin, C.: On the ambiguous interpretation of the Turc-Budyko nondimensional graph, Water Resources Research, 48, https://doi.org/10.1029/2012wr012532, 2012.

Ballabio, C., Panagos, P., and Monatanarella, L.: Mapping topsoil physical properties at European scale using the LUCAS database, Geoderma, 261, 110–123, https://doi.org/10.1016/j.geoderma.2015.07.006, 2016.

Ebbinghaus, H.: Über das gedächtnis: untersuchungen zur experimentellen psychologie, Duncker & Humblot, 1885.

Iliopoulou, T., Aguilar, C., Arheimer, B., Bermúdez, M., Bezak, N., Ficchì, A., Koutsoyiannis, D., Parajka, J., Polo, M. J., Thirel, G., and Montanari, A.: A large sample analysis of European rivers on seasonal river flow correlation and its physical drivers, Hydrology and Earth System Sciences, 23, 73–91, https://doi.org/10.5194/hess-23-73-2019, 2019.

Klemeš, V., Srikanthan, R., and McMahon, T. A.: Long-memory flow models in reservoir analysis: What is their practical value?, Water Resources Research, 17, 737–751, https://doi.org/10.1029/wr017i003p00737, 1981.

McDonnell, J. J.: Beyond the water balance, Nature Geoscience, 10, 396–396, https://doi.org/10.1038/ngeo2964, 2017.

Oudin, L., Hervieu, F., Michel, C., Perrin, C., Andréassian, V., Anctil, F., and Loumagne, C.: Which potential evapotranspiration input for a lumped rainfall–runoff model? Part 2 – Towards a simple and efficient potential evapotranspiration model for rainfall–runoff modelling, Journal of Hydrology, 303, 290–306, https://doi.org/10.1016/j.jhydrol.2004.08.026, 2005.

Pelletier, A. and Andréassian, V.: Hydrograph separation: an impartial parametrisation for an imperfect method, Hydrology and Earth System Sciences, 24, 1171–1187, https://doi.org/10.5194/hess-24-1171-2020, 2020.

Pelletier, A., Andréassian, V., and Delaigue, O.: baseflow: Computes Hydrograph Separation, https://doi.org/10.15454/Z9IK5N, r package version 0.13.2, 2021.

Risbey, J. S. and Entekhabi, D.: Observed Sacramento Basin streamflow response to precipitation and temperature changes and its relevance to climate impact studies, Journal of Hydrology, 184, 209–223, https://doi.org/10.1016/0022-1694(95)02984-2, 1996.

Staudinger, M., Stoelzle, M., Seeger, S., Seibert, J., Weiler, M., and Stahl, K.: Catchment water storage variation with elevation, Hydrological Processes, 31, 2000–2015, https://doi.org/10.1002/hyp.11158, 2017.

---

## Author Response (AR2)

**Second answer to reviewers**

**Quantifying pluriannual hydrological memory with Catchment Forgetting Curves**

Alban de Lavenne[1,2], Vazken Andréassian[2], Louise Crochemore[1,3], Göran Lindström[1], and Berit Arheimer[1]

[1]SMHI, Norrköping, Sweden
[2]Université Paris-Saclay, INRAE, UR HYCAR, Antony, France
[3]INRAE, UR RiverLy, Lyon, France

**Correspondence:** Alban de Lavenne (alban.delavenne@inrae.fr)

**Report #1**

> I think the authors did a decent job in revising some details of the paper. The paper seems mostly ready for publication. However, please check the minor suggestions below in submitting the final version.

Thanks for the additional comments on this second version and for the appreciation of the work done. We have addressed your suggestions below.

> **Abstract**
>
> 5      A climatic anomaly can potentially affect **on** the hydrological behaviour of a catchment for several years. Remove ""on".

"on" has been removed.

> I think elasticities "quantifies" (i.e. not "measures").

We now use "quantifies".

> CFCs are parameterized using a Gamma distribution derived from the calibration of Gamma distributions. Without context what these gamma distributions represent, this is hard to understand for a reader.

The sentence was removed from the abstract.

> L10 It would help to specify is that was your expectation.

Our expectations are specified in the following sentence : "As expected, French catchments overlying large aquifers exhibit a long memory, i.e. with the impact of climate anomalies detected over several years."

> L15-16 "Our work thus15 underlines the need to account for catchment memory in order to produce meaningful and geographically coherent elasticity indices." is not really supported since for 85% of the catchment the method did not improve anything significantly?

It's true that only 15% of our catchments have significant multi-year memory (as now stated in the abstract). However, it's enough to produce some irrelevant spatial patterns of elasticity indices, as highlighted by Figure 9, if this memory is not taken into account.

> **Introduction**
>
> L24 "fora Turc-Budyky explanation framework" is awkwardly phrased. Do you mean for an explanation o the Turc-Budyko framework. However, before revising this sentence. What does this Turc-Budyko framework have to do with runoff response given the antecedent catchment wetness. The wetness described in such a framework is the climatic wetness (or aridity) that does not directly describe the state of soils (but only indirectly relates).

We deleted the sentence about soil conditions here, to actually be closer to the moisture defined in the Turc-Budyko framework. We now say: "(The response of a catchment to incoming precipitation depends largely on its *wetness* (see e.g. Andréassian and Perrin (2012) for an explanation within the Turc-Budyko framework)."

> I remain to struggle with understanding why section 1.2 needs to be this long, and not to the point. By containing unnecessary much information about water age versus catchment hydrometric response, the reader can be strongly distracted from the main contribution of this paper. I would strongly recommend to drop the part on "water age" and just state in a single sentence that this work focusses "on quantifying the response of catchments in terms of flow volume, independent of the water age of this flow."

We do not agree on this point. We believe that this is an important distinction to make and that this idea cannot be summarized in a single sentence, and for this reason it deserves a full sub-section (in the previous review, referee #2 agreed on this aspect): We really want to avoid the CFC being too quickly interpreted as a travel time distribution, which could be quite tempting at first because both notions have a temporal aspect.

> The new addition "This distinction may also be linked to the different perceptual storages of water in a catchment. "Mobile storage", which controls transport in a catchment is more linked to water age, whereas "dynamic storage" which controls streamflow dynamics is more in line with our definition of catchment memory (see e.g. Staudinger et al., 2017)" is a rather unclear description of something that seems not to be directly relevant to the paper.

This new reference was suggested by referee #2, and we agreed on the relevance of this addition. Similarly to the previous comment, we think that this section is necessary to avoid any misunderstanding of what we quantify in this study. This reference helps us to introduce a vocabulary that can be found in other papers, in order to help the reader establish relevant links to these studies that follow a very close conceptualisation of the hydrological response.

> L289: "It thus appears that the hydrological behaviour of the driest catchments is more dependent on past climatic conditions than that of humid catchments". Please rephrase this sentence as right now it makes an wrong comparison.

We now say: "It thus appears that past climatic conditions have more influence on the hydrological behaviour of the driest catchments than on that of humid catchments."

> L388: break this into two sentences.

We now say: "Future work could also design dynamic CFCs by investigating how climate anomalies might change the shape of CFCs over time."

**Report #2**

> I gave a fresh read to the revised paper, and I think it addresses most issues of the previous submission.

40    Thanks for the additional comments on this second version and for the appreciation of the work done. We have addressed your suggestions below.

> Some additional points to consider are the following:
>
> Introduction. About water age, the authors write "Because we do not use any tracers in this study, we cannot check any hypothesis about water age and we will not discuss this topic further". But it is not really true that the topic is not discussed further, as below they write: "Spectral analysis can be used to provide insight into catchment memory. It is regularly used for stream chemistry (see e.g. Kirchner et al., 2000), in order to understand travel time distributions.", and again "has a long history when it comes to water quality modeling or tracer analysis, as past pollution inputs can influence water quality in rivers for several years or decades". Please rephrase.

We have rephrased the sentence as follows: "Because we do not use any tracers in this study, we cannot check any hypothesis about water age and we will not provide any related interpretations in our analysis afterwards;"

> By reading the review, it appears that many past studies used the terminology "catchment memory". It should be
45    made clear that this is the terminology used in the current study, and adopted to describe analogous concepts in earlier work.

We fully agree that many past studies used the same terminology "catchment memory", and this is what our introduction aims to review. We tried to make that clearer by saying: "In addition to the "memory" terminology, the scientific literature sometimes addresses this concept also through "flow persistence" (see e.g., Svensson, 2015; Quinn et al., 2021) or "flow predictability" (see e.g., Bierkens and van Beek, 2009; van Dijk et al., 2013)."

> 50    Line 209: Figure 1 shows an example... It should be Figure 1b (or refer to panel b).

We now refer to Figure 1a and Figure 1b specifically.

> Methods should present how elasticity and memory results from different catchments are going to be assessed. Such as, that they will be interpreted on the basis of catchment attributes, and which catchment attributes are going to be used. Otherwise this only appears in the results. Hence, methods should present and motivate the "few hydroclimatic characteristics commonly identified as the main drivers in the literature".

Catchment descriptors were already presented in section 2.1. We moved them to a different section called "Identification of hydro-climatic drivers of catchment memory" at the end of section 2 where we now present in more detail where they come from and how they are used to understand the memory drivers.

> I find that having here having the results and discussion section combined is a bit problematic, because one needs the overall results picture because making interpretations. For example, statements such as "Our conclusion is that catchment size is not a first-order determining factor of memory and elasticity, and this likely reflects some more regional relation between catchment size and hydrology" would be more appropriate in the synthesis section. I therefore suggest to limit the interpretation to the minimum necessary in the results section, and expand the synthesis section (currently very short), to provide a comprehensive critical elaboration of the results. Otherwise, the risk is that the main message of the paper will be relegated to the last paragraph of the synthesis section, which provides a summary, but with no interpretation.

We deliberately made a short synthesis that can provide the main *take-home messages* of the paper. This statement about the effect of catchment size is actually already mentioned in this synthesis section: "Catchment area, often referred to in the literature, does not seem to play a first-order role". However, we have extended the hydrological interpretation of the results in this section.

**References**

Andréassian, V. and Perrin, C.: On the ambiguous interpretation of the Turc-Budyko nondimensional graph, Water Resources Research, 48, https://doi.org/10.1029/2012wr012532, 2012.

Bierkens, M. F. P. and van Beek, L. P. H.: Seasonal Predictability of European Discharge: NAO and Hydrological Response Time, Journal of Hydrometeorology, 10, 953–968, https://doi.org/10.1175/2009jhm1034.1, 2009.

Quinn, D. F., Murphy, C., Wilby, R. L., Matthews, T., Broderick, C., Golian, S., Donegan, S., and Harrigan, S.: Benchmarking seasonal forecasting skill using river flow persistence in Irish catchments, Hydrological Sciences Journal, https://doi.org/10.1080/02626667.2021.1874612, 2021.

Svensson, C.: Seasonal river flow forecasts for the United Kingdom using persistence and historical analogues, Hydrological Sciences Journal, 61, 19–35, https://doi.org/10.1080/02626667.2014.992788, 2015.

van Dijk, A. I. J. M., Peña-Arancibia, J. L., Wood, E. F., Sheffield, J., and Beck, H. E.: Global analysis of seasonal streamflow predictability using an ensemble prediction system and observations from 6192 small catchments worldwide, Water Resources Research, 49, 2729–2746, https://doi.org/10.1002/wrcr.20251, 2013.